# Effects of Nonlinearity on Velocity, Acceleration and Pressure Gradient in Free-Stream Zone of Solitary Wave over Horizontal Bed—An Experimental Study

**Chang Lin [1],\*, Ming-Jer Kao [1], James Yang [2,3] , Juan-Ming Yuan [4] and Shih-Chun Hsieh [5]**

[1] Department of Civil Engineering, National Chung Hsing University, Taichung 40227, Taiwan
[2] Vattenfall AB, R&D Hydraulic Laboratory, 81426 Älvkarleby, Sweden
[3] Civil and Architectural Engineering, KTH Royal Institute of Technology, 10044 Stockholm, Sweden
[4] Department of Data Science and Big Data Analytics, Providence University, Taichung 43301, Taiwan
[5] Axesea Engineering Technology Co., Ltd., Taichung 407034, Taiwan
\* Correspondence: chenglin@nchu.edu.tw

**Abstract:** For solitary waves on a horizontal bed, the study deals experimentally with the high ratio of wave height ($H_0$) to still water depth ($h_0$) that amplifies the wave nonlinearity. The value of $H_0/h_0$ tested in a wave flume ranges from 0.050 to 0.550, indicating the shift from a quasi-linear solitary wave to a highly nonlinear one. A high-speed particle image velocimetry (HSPIV) and a flow visualization technique of particle-trajectory tracking method are utilized to measure velocity fields and identify near-bed flow structures. The unsteady free-stream velocities with equal magnitude take place in a free-stream zone, FSZ). The FSZ underlies the internal flow zone, over which the external free surface of solitary wave exists and is situated beyond the boundary layer. The spatio-temporal variation of free-stream velocity, moving in phase with the free surface elevation, characterizes the pattern of pressure gradient in the FSZ and thus dominates the behavior of boundary layer flow. Accordingly, nonlinear effects on the time series as well as the maximum values of horizontal velocity, particle acceleration, and pressure gradient in the FSZs of solitary waves are presented. Before, at, and after the wave crest's intersection with a given measurement location, favorable, zero, and adverse pressure gradients occur in the FSZ, respectively. For $H_0/h_0 = 0.179$, 0.363, and 0.550, the values of the dimensionless maximum free-stream velocity are about 3.10, 5.32, and 6.20 times that (= 0.0473) for $H_0/h_0 = 0.050$; and the corresponding values of the dimensionless maximum adverse pressure gradient are about 5.74, 14.54 and 19.84 times that (= 0.0061) for $H_0/h_0 = 0.050$. This evidence highlights the nonlinear effect on the kinematic and hydrodynamic features of solitary waves. Finally, the effect of nonlinearity on the relationship between the dimensionless time for the maximum adverse pressure gradient in the FSZ and that for the incipient flow reversal in the bottom boundary layer is explored for the first time. It is found that the incipient flow reversal takes place immediately after the maximum adverse pressure gradient, together with a decrease in the dimensionless time for flow reversal if $H_0/h_0$ increases. The fact accentuates the nonlinear effect on the incipient flow reversal right above the bed.

**Keywords:** solitary wave; nonlinearity; free-stream velocity; acceleration; pressure gradient; incipient flow reversal

## 1. Introduction

The movement of a solitary wave was first observed in situ by Russell [1] with quite a stable waveform traveling over a long distance. His experimental results indicated that the wave propagated at a nearly constant wavelength (or celerity) yet a small attenuation in wave height [2], thus recognized as one type of permanent wave. A solitary wave is also regarded as one type of long-wave because the wave propagates persistently over shallow water. As reported by Grue et al. [3] and Madsen et al. [4], a combination of

multiple solitary waves with distinct wave heights and different separation times was observed in the field during tsunami events. Further, El et al. [5] and Grilli et al. [6] demonstrated both numerically and experimentally that a series of isolated leading solitary waves emerged while an undular bore traveled at a decreasing water depth. Based on the illustrations mentioned above, a single solitary wave or multiple solitary waves take place in the navigation (or irrigation) channel, estuary, coastal zone, or ocean strait. The fact highlights the importance of associated studies on solitary waves.

It is known that multiple approaches are employed to predict the free surface elevation and kinematic features of a solitary wave traveling on a horizontal bed. These include the analytic and theoretical solutions (Boussinesq [7], McCowan [8], Munk [9], Synolakis [10], Liu, et al. [11], and Gavrilyuk et al. [12]), the higher-order approximations (e.g., Grimshaw [13] and Fenton [14]), and simulations with CFD and incompressible smoothed particle hydrodynamics (ISPH) (e.g., Higuera et al. [15] and Aly [16]).

Limited experimental investigations focus on the free surface elevations and particle velocities of solitary waves over a horizontal bed (Lee et al. [17] and Lin et al. [18–20]). Employing a resistance wave gauge and laser Doppler velocimetry, Lee et al. [17] measured the time series of free surface elevations as well as the counterparts of horizontal and vertical velocities for solitary waves with a ratio $H_0/h_0 = 0.11$–$0.29$. Comparisons of the experimental results with those predicted by Boussinesq, McCowan, Munk, and Grimshaw solutions were then made, aiming to examine the validity of these analytic solutions. Comparisons of the measured free surface elevations with the predicted ones by the four solutions all showed good agreement. However, comparisons between the measured horizontal and vertical velocities and the predicted ones indicated that the Boussinesq solution matched satisfactorily with the experimental data trend. Note that the nonlinear effect of solitary waves was not discussed in their study.

Using the HSPIV, Lin et al. [18] demonstrated the "similarity profiles" for the dimensionless horizontal velocity; and for the dimensionless time lead of horizontal velocities in the boundary layer flows induced by solitary waves traveling over a horizontal bed with $H_0/h_0 = 0.096$–$0.386$. Their study lacked in exploring the nonlinear effect on the kinematic and hydrodynamic features. As reported by Lin et al. [19,20], the features of (local and convective) accelerations and pressure gradient all over the internal flow as well as the relevant "similarities and Froude number similitudes" were elucidated only for a solitary wave (with $H_0/h_0 = 0.363$) propagating on a horizontal bed. Further, as indicated by Lin et al. [20], fairly good predictions of horizontal and vertical velocities could be obtained by the Boussinesq solution if the linear wave celerity is employed in the computation. A series of studies on kinematic and hydrodynamic features or flow similarity and Froude number similitude was reported in Lin et al. [21–26] for the run-up and run-down motions of solitary waves traveling over 1:15–1:3 sloping beaches. Recently, Lin et al. [27] studied the characteristics of flow velocity and pressure gradient of an undular bore propagating on a horizontal bed.

Based on the above literature survey, it is still rudimentary concerning the role that the nonlinear effects play in the relevant kinematic and hydrodynamic features of solitary waves over a horizontal bed. In other words, as influenced by the nonlinearity, the corresponding features of flow velocities, particle accelerations, and pressure gradients of solitary waves are not made clear. Further, the relationship between the time instants for the occurrence of the maximum adverse pressure gradient in the FSZ and the incipient flow reversal in the boundary layer has not been investigated. Thus, using an HSPIV and a flow visualization technique, this study focuses on these unknown kinematic and hydrodynamic features, aiming to enhance the understanding of long-wave mechanics.

The outline of this paper is as follows. Section 2 illustrates the experimental set-up and instrumentation, followed by Section 3 with the validation tests. The results and discussions are presented in Section 4. Finally, the findings are concluded in Section 5.

## 2. Experimental Set-Up and Instrumentation

### 2.1. Wave Flume and Coordinate System

Experiments were performed in a wave flume (14.00 m long, 0.25 m wide, and 0.50 m high) at the Department of Civil Engineering, National Chung Hsing University, Taiwan. A piston-type wave maker (WM-PT-S, Chen Hseng Machine Co., Ltd., Taichung, Taiwan, ROC), placed at one end of the flume and had a maximum stroke of 49.0 cm, was actuated by a servomotor to follow the wave-plate trajectory (Goring [28]) and produce a solitary wave. As reported by Lin et al. [15,18–26], the generation of a satisfactory solitary wave by the wave maker was highly repeatable, together with the dispersive tail-wave train almost indiscernible.

A Cartesian coordinate system, $(x, y)$, is set up with its origin $(0, 0)$ on the horizontal bed surface (Figure 1). Axis $x$ is along the flume and axis $y$ is upward from the surface of the bed. The specified measuring section (SMS), $x = 0$, is positioned 800.0 cm from the wave maker at rest. Let $t$ denote time, $t_s = (h_0/g)^{1/2}$ a representative time scale with $g$ being the gravity acceleration, thus leading to the dimensionless time, $T [= t/t_s = t \times (g/h_0)^{1/2}]$. It should be mentioned that $t = 0$ specifies the time instant when the wave crest passes the SMS. The velocity components, $(u, v) = (u[x, y, t], v[x, y, t])$, represent the horizontal and vertical velocities of water particles along the $(x, y)$ directions, respectively.

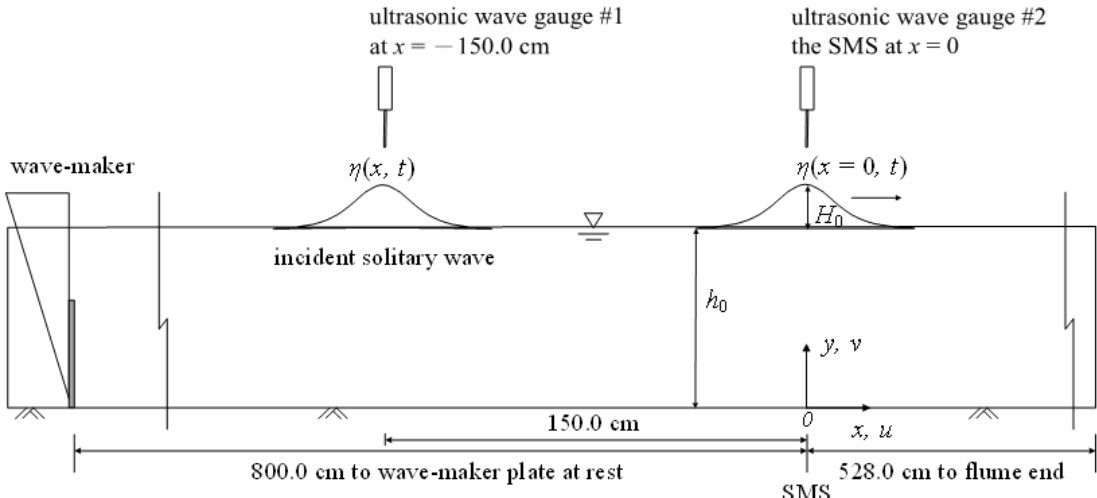

**Figure 1.** Schematic diagram showing a solitary wave propagating over a horizontal bed, two wave gauges, the $(x, y)$ coordinate system, and $(u, v)$ velocity components.

### 2.2. Wave Gauge, HSPIV, and Flow Observation

As shown in Figure 1, two ultrasonic wave gauges (Banner U-Gage S18U, Banner Engineering Corp., Tenth Avenue North, MN, USA) were placed at $x = -150.0$ cm and the SMS measured the free surface elevations, $\eta(x, t)$. The former was utilized as a reference signal to trigger the start of data sampling when it reached the preset threshold (i.e., a specified voltage somewhat smaller than the one corresponding to the wave crest). The latter obtained the temporal variation of free surface elevation, $\eta(0, t)$, and then determined $H_0$ at the SMS.

Instantaneous velocity fields were measured by an HSPIV system, comprising a high-speed digital camera and a laser head. A 5 W argon-ion laser (Coherent Innova-90, Santa Clara, CA, USA) was the light source. A cylindrical lens expanded the laser beam from the laser head to form a 1.5 mm thick fan-shaped light sheet, which was then directed upwards via the bottom glass along the half span of the wave flume. Titanium dioxide particles (mean diameter 1.8 μm) were uniformly seeded into the water volume in the flume. The digital camera (Phantom VEO640 or Phantom V5.1, Vision Research Inc., Wayne, NJ, USA), with a maximum framing rate of 500/1000/1600–2500 Hz at a resolution of (1152 × 1152)/(1024 × 512)/(2048 × 1152) pixel, captured the instantaneous particle-

laden images. Before commencing the cross-correlation computation for the instantaneous velocity fields, the Laplacian edge-enhancement technique (Adrain and Westerweel [29]) and contrast enhancement method (Cowen and Monismith [30]) were utilized to brighten the traced particles and sharpen the particle edges in the images. The HSPIV algorithm allowed the instantaneous velocity field to be computed from a pair of images, starting with an interrogation window from $64 \times 64$ pixels and ending at $8 \times 8$ pixels.

This study aims to reveal the effect of the nonlinearity of solitary waves on horizontal velocities, particle accelerations, and pressure gradients in the FSZs, located in the upper portions of near-bottom zones. Therefore, the fields of view of the camera in velocity measurements (and also for instantaneous flow observation) by the HSPIV were set with distinct sizes as listed in Table 1, and their centers were located at the SMS. The framing rate of the camera was set at 1600–2500 or 1000 or 500 Hz, depending on the experimental conditions. Note that the images captured by the HSPIV also provide the information of instantaneous flow field with the "*particle-dotted*" pattern, which is beneficial to precisely identify the time for the occurrence of the incipient flow reversal (i.e., water particles starting to move opposite to the wave propagation direction) exactly above the bed. A total of 20 repeated runs for the velocity measurements were performed for each case.

**Table 1.** A list of experimental conditions.

| Case | $H_0$ (cm) | $H_0/h_0$ * | $C$ (cm/s) | $C_0$ (cm/s) | $C_0/C$ | Framing Rate of HSPIV (Hz) | Framing Rate of FV (Hz) | Size (cm × cm) (Length × Width) |
|------|-----------|-------------|------------|--------------|---------|----------------------------|-------------------------|----------------------------------|
| A | 0.40 | 0.050 | 88.59 | 90.78 | 1.025 | 1600 | 30 | 2.05 × 1.15 |
| B | 0.90 | 0.112 | 88.59 | 93.44 | 1.055 | 2500 | 50 | 2.05 × 1.15 |
| C | 1.43 | 0.179 | 88.59 | 96.18 | 1.086 | 2500 | 50 | 2.05 × 1.15 |
| D | 2.28 | 0.285 | 88.59 | 100.42 | 1.134 | 2500 | 100 | 2.05 × 1.15 |
| | | | | | | 500 | - | 16.80 × 16.80 (HSPIV) |
| E | 2.90 | 0.363 | 88.59 | 103.41 | 1.167 | 1000 | - | 2.00 × 1.00 (HSPIV) |
| | | | | | | - | 100 | 2.05 × 1.15 (FV) |
| F | 3.08 | 0.385 | 88.59 | 104.26 | 1.177 | 2500 | 100 | 2.05 × 1.15 |
| G | 3.52 | 0.440 | 88.59 | 106.31 | 1.200 | 2500 | 100 | 2.11 × 1.18 |
| H | 4.00 | 0.500 | 88.59 | 108.50 | 1.225 | 2000 | - | 2.11 × 1.18 |
| I | 4.40 | 0.550 | 88.59 | 110.29 | 1.245 | 2500 | 100 | 2.11 × 1.18 |

* Note that $h_0 = 8.0$ cm for all cases.

The particle-trajectory tracking method helped visualize the flow structure in the near-bottom zone, with the purpose to observe the commencement of flow reversal right beyond the surface of the horizontal bed. The camera, laser light sheet, and seeding particles used for this technique were all the same as those for velocity measurements by the HSPIV. The fields of view for flow observations had a size of (2.00–2.11) cm in length and (1.00–1.18) cm in height, centered at the SMS. For each case, a total of three repeated runs were carried out to capture the flow fields characterized by the "*pathlined*" pattern. The sampling rates of the camera were set at 30–100 Hz, depending on the experimental conditions (see Table 1).

### 2.3. Experimental Conditions

Nine experimental cases were tested with $H_0$ in the range 0.4–4.4 cm but at the same $h_0 = 8.0$ cm, thus $H_0/h_0 = 0.05$–0.55 (Cases A–I). The ratio reflects a solitary wave's transition from a quasi-linear state to a highly nonlinear one [11,15,17,18,23–25]. To acquire the time series of *instantaneous* horizontal velocity in the free-stream zone at the SMS, a symmetric 11-point smoothing scheme with distinct weightings was used to eliminate noises in the velocity data. The ensemble-average method was then employed for the 20 repeated runs to obtain the time series of *ensemble-averaged* horizontal velocity. Table 1 summarizes the nine experimental conditions. For all cases, the values of the linear and nonlinear wave celerity, $C$ [$= (gh_0)^{1/2}$] and $C_0$ ($= [g(h_0 + H_0)]^{1/2}$), as well as the counterparts of $C_0/C$ are listed in Table 1. The relationship between $C_0/C$ and $H_0/h_0$ is regressed to be $C_0/C = (0.4546H_0/h_0 + 1.0)$. For Case E, only the flow observation using particle-trajectory

tracking was conducted without performing velocity measurement by the HSPIV. These un-done counterparts were previously made throughout the full water depth and partially reported in Lin et al. [18–20]. In this study, elucidation of the kinematic and hydrodynamic features for Case E was achieved by mining the data banks that were set up during the experimental stage of the three studies.

## 3. Validation Tests

Figure 2 presents a comparison of the original time series of free surface elevation $\eta_o(t)$ for five runs, which are randomly selected from a total of 20 runs for Case E. Their profiles of the time series almost collapse onto each other without prominent discernible distinction. The fact still holds true for the other eight experimental cases. The evidence demonstrates the high repeatability of the solitary waves generated by the wave maker. A comparison of the time series of ensemble-averaged free surface elevation $\eta(t)$ (i.e., obtained by phase-averaging over 20 repeated runs) with those predicted by solitary wave theory [31] is illustrated in Figure 3 for Case E. The trend of measured data in $\eta(t)$ is in good accordance with that predicted by solitary wave theory.

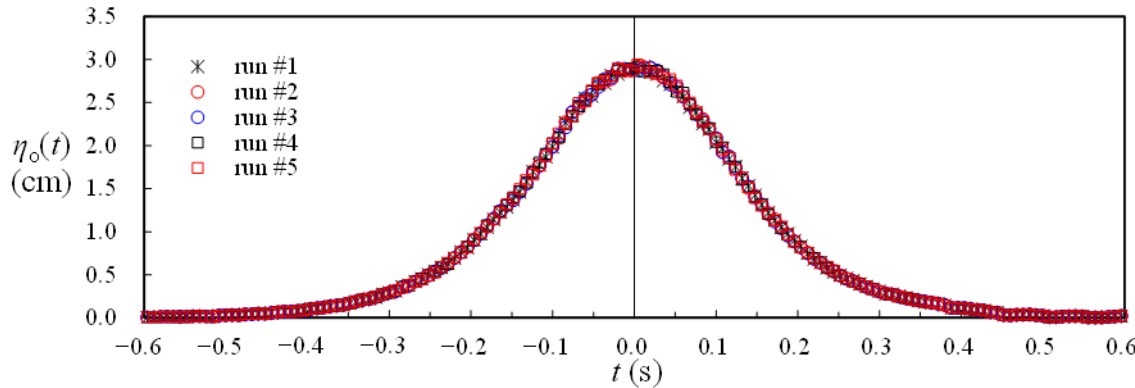

**Figure 2.** A comparison among five repeated runs for the original time series of free surface elevation $\eta_o(t)$ (Case E).

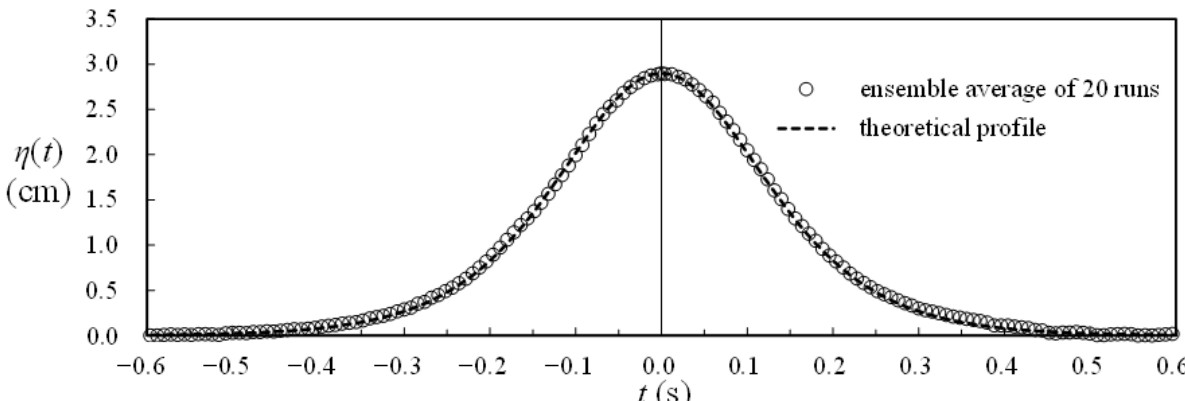

**Figure 3.** A comparison of time series between ensemble-averaged free surface elevation $\eta(t)$ and predicted one (Case E).

Further, for Case E, Figure 4a,b demonstrates two ensemble-averaged velocity fields for 1.0 cm $\leq y \leq$ 8.0 cm at $t = -0.045$ and 0.045 s, respectively. The measurement errors of these velocities, obtained by using HSPIV, are estimated herein by the mass flux method (Chang and Liu [32]). Based on the two-dimensional flow field, a check is first done by calculating the mass flux of each measuring grid, $M_f = |\partial u/\partial x + \partial v/\partial y| \times dA$. Note that, due to mass conservation and flow continuity, $M_f$ should be equal to zero theoretically. For

the velocity field for Case E, the grid lengths are $\Delta x = \Delta y = 0.1167$ cm, and the corresponding area is $dA = \Delta x \times \Delta y = 0.0136$ cm$^2$. A representative mass flux, $M_{fr}$ [= $(u_{wc})_{max} \times \Delta y$], is designated with $(u_{wc})_{max}$ [= 36.94 cm/s] being the maximum horizontal velocity at wave crest for $T = 0$. The relative error is then defined by $M_f/M_{fr}$. The values of $M_f/M_{fr}$ are found to be below 2.0%. Moreover, for a typical horizontal velocity of 20.0 or 40.0 cm/s in the flow field, the measurement error of horizontal velocity is about 0.3 or 0.46 cm/s (Lin et al. [18,22,24]), rendering the relative error to be 1.5% or 1.2%. These two pieces of evidence do indicate high precision in the velocity measurements utilizing the HSPIV system.

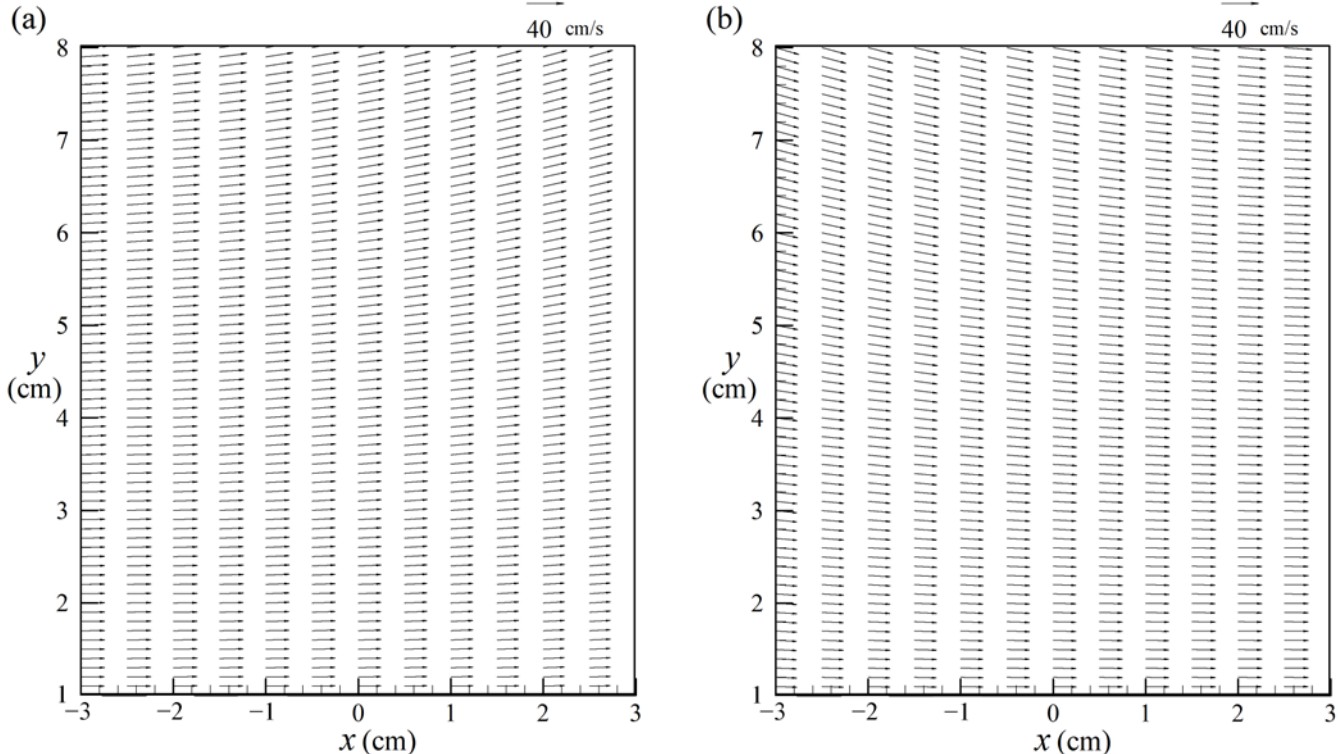

**Figure 4.** The (ensemble-averaged) velocity fields obtained at (**a**) $t = -0.045$ s; (**b**) $t = 0.045$ s (Case E).

As pointed out by Lee et al. [17] and Lin et al. [20], the Boussinesq solution incorporated with $C$ in the calculation provides a good prediction of flow velocity. For Case E, Figure 5a–c show comparisons of the ensemble-averaged horizontal velocities $u(t)$ obtained at $y = 7.04$, 4.40, and 1.44 cm, respectively, with those predicted by the Boussinesq solution. Good accordance between the experimental data and the theoretic predictions is evidenced, reconfirming the satisfactorily precise velocity measurements by the HSPIV system. Moreover, Figure 6a–c presents the comparisons between the time series of local acceleration $A_1(t)$ (calculated from the time series of the ensemble-averaged horizontal velocity) and those predicted by wave theory at $y = 7.04$, 4.40, and 1.44 cm. Note that the details of the calculation procedure for the former are addressed later in Sec. 4.3, and the latter is directly computed by taking the time derivative of the horizontal velocity predicted by the Boussinesq solution. Not only for the entire trend but also for the individual magnitudes in these time series, fairly good agreement is thus confirmed.

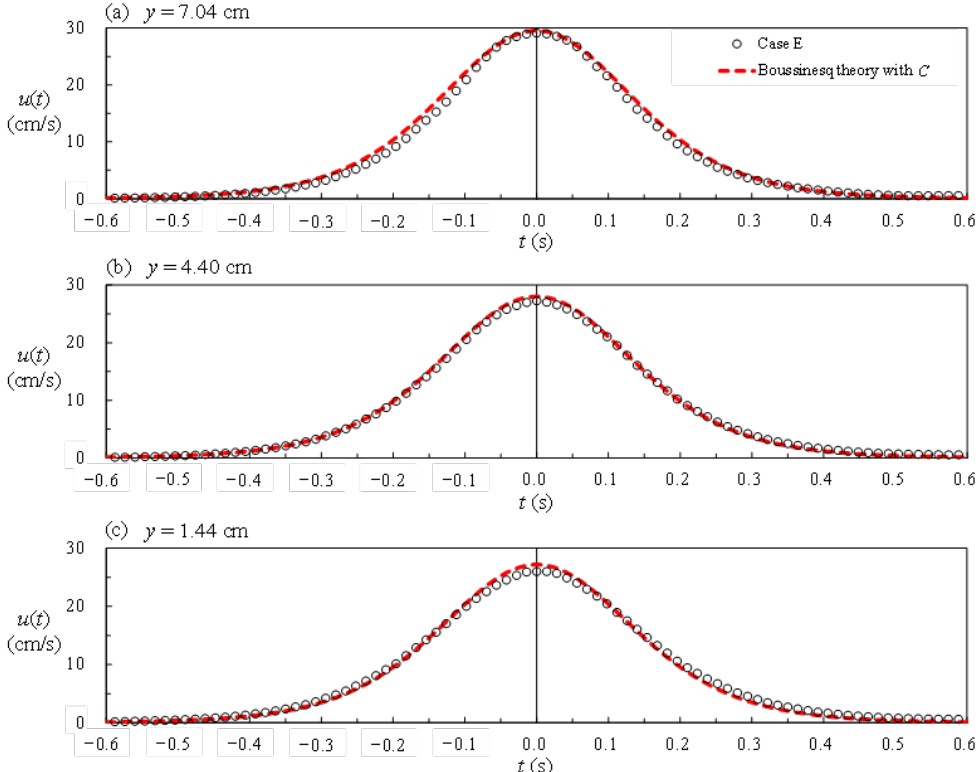

**Figure 5.** Time series comparisons of horizontal velocity between measured and predicted using Boussinesq solution (Case E) at $y$ = (**a**) 7.04 cm; (**b**) 4.40 cm; and (**c**) 1.44 cm.

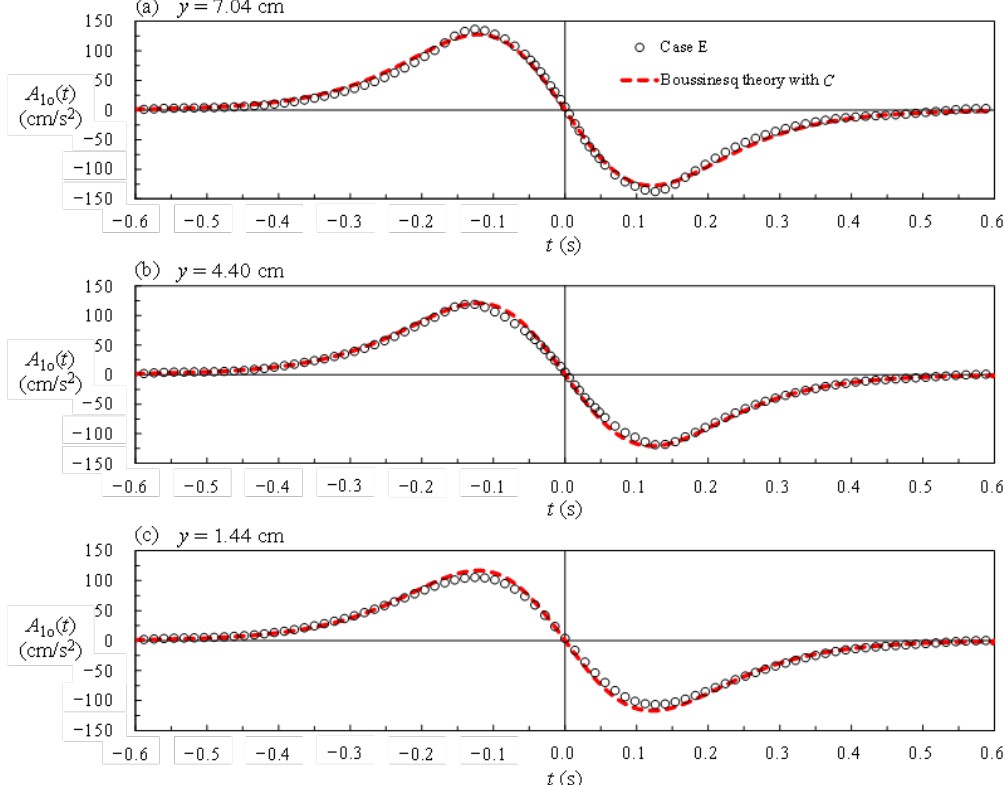

**Figure 6.** Time series comparisons between calculated local accelerations (which are based on time series of measured horizontal velocities) and predicted ones using Boussinesq solution (Case E) at $y$ = (**a**) 7.04 cm; (**b**) 4.40 cm; and (**c**) 1.44 cm.

## 4. Results and Discussion

A series of experimental results of geometric, kinematic, and hydrodynamic features with dimensionless forms are presented in the following. The representative length and time scales considered are $h_0$ or $[h_0 + H_0]$ (or simply $H_0$) and $(h_0/g)^{1/2}$ or $[(h_0 + H_0)/g]^{1/2}$. The two scales result in the representative velocity scale being $C = (gh_0)^{1/2}$ or $C_0$ (= $[g(h_0 + H_0)]^{1/2}$), which is equivalent to the linear or nonlinear wave celerity. The length and time scales selected herein are $h_0$ and $(h_0/g)^{1/2}$, together with the use of $H_0$ to normalize the time series of free surface elevation, $\eta(T)$. To highlight the nonlinear effect, the nonlinear wave celerity, $C_0$ (= $[g(h_0 + H_0)]^{1/2}$), is particularly used as the velocity scale. Note that $C_0/C$, =; $(04546H_0/h_0, +; 10)$; as evidenced in Table 1, thus allowing prompt transformation between $C_0$ and $C$. Further, the acceleration (or pressure gradient) scale amounts to the ratio of the velocity scale to the time scale, i.e., equal to $g$.

### 4.1. Elucidation of FSZs from Velocity Profiles

To show the presence of the FSZ beyond the boundary layer and underlying the internal zone, it is appropriate to first use the velocity profiles all over the full water depth. For Case E at the SMS ($x/h_0 = 0$), Figure 7a,b–h present the time series of dimensionless free surface elevation, $\eta(T)/H_0$, as well as the profiles of dimensionless horizontal and vertical velocities, $u(y/h_0, T)/C_0$ and $v(y/h_0, T)/C_0$, at $T = -2.50, -1.39, -0.50, 0, 0.5, 1.39,$ and $2.50$ (as marked by dashed lines with ①−⑦ in Figure 7a), respectively. The times before, at, and after the intersection of the wave crest with the SMS correspond to $-6.00 \le T < 0$, $T = 0$, and $0 < T \le 6.00$, respectively, with an ascending, constant, and descending free surface. With reference to $T = 0$, the free surface elevations for $-6.00 \le T < 0$ are symmetric to those for $0 < T \le 6.00$, exhibiting an even-function form with $\eta(T)/H_0 = \eta(-T)/H_0$. In addition, as evidenced in Figure 7b–h, the horizontal velocities are positive with $u(y/h_0, T)/C_0 = u(y/h_0, -T)/C_0$ throughout the full water depth at the SMS. However, the vertical velocities are negative, zero, and positive with $v(y/h_0, T)/C_0 = -v(y/h_0, -T)/C_0$, except for those very close to the bed.

As shown in Figure 7b or Figure 7h for $T = -2.50$ or $2.50$, $u(y/h_0)/C_0$ increases from zero at $y/h_0 = 0$ to $0.082$ at $y/h_0 = 0.022$ or $0.031$, characterizing the feature of the boundary layer. It then remains almost constant for $y/h_0 = 0.031$–$0.35$, as indicated by the two horizontal dotted lines within which the uniform horizontal velocities exist. This layer is herein termed the FSZ within which the horizontal velocities are denoted as the *free-stream velocities*. For $y/h_0 > 0.35$, $u(y/h_0)/C_0$ decreases linearly to $0.066$ at $y/h_0 = y_{fs}/h_0 = 1.08$ (i.e., the instantaneous free surface). It is worth mentioning that, for $-6.00 \le T < -1.39$ or $1.39 < T \le 6.00$ and $y/h_0 > 0.35$, all the dimensionless horizontal velocity profiles exhibit similar shapes to that at $T = -2.50$ or $2.50$. In particular, the upper portions have a non-uniform, linearly decreasing trend. This differs from the known recognition of long waves having a uniform horizontal velocity profile all over the full water depth. As seen in Figure 7c or Figure 7g at $T = -1.39$ or $1.39$ (i.e., at phase corresponding to the "inflection point" in Figure 7a), $u(y/h_0)/C_0$ ranges from zero at the bed to $0.163$ at $y/h_0 = 0.024$ or $0.029$, then remains nearly unchanged from the lower limit of FSZ (at $y/h_0 = 0.03$) to the free surface.

Further, for $-1.39 < T \le 0$ or $0 \le T < 1.39$ (Figure 7d,e or Figure 7e,f), the free-stream velocity prevails at $y/h_0 = 0.026$–$0.35$ or $0.027$–$0.35$. Especially, for $y/h_0 > 0.35$, a distribution with a nonlinear increase in the horizontal velocity is evidenced. The fact is again contrary to the uniform horizontal profile along the full water depth, assumed frequently in the theoretical prediction or numerical simulation of long-wave propagation. The maximum values of dimensionless horizontal velocity, $[u(y/h_0, T)]_{max}/C_0$, occur at $T = 0$ with the FSZ located at $y/h_0 = 0.027$–$0.35$. Note that the extreme value of $[u(y/h_0, 0)]_{max}/C_0$, taking place right at the wave crest (i.e., $y/h_0 = y_{crest}/h_0 = 1.363$), is equal to $0.357$, greater than those (= $0.251$) in the FSZ.

As also shown in Figure 7b–d for $-6.00 \le T < 0$ or Figure 7f–h for $0 < T \le 6.00$, the dimensionless vertical velocity in each profile, $v(y/h_0, T)/C_0$, linearly varies from

zero at the bed to a certain positive or negative maximum, $[v(y_{fs}/h_0, T)]_{max+}/C_0$ or $[v(y_{fs}/h_0, T)]_{max-}/C_0$, at the instantaneous free surface. The profile with all positive or negative maxima, $[v(y/h_0)]_{max+}/C_0$ or $[v(y/h_0)]_{max-}/C_0 \; (= -[v(y/h_0)]_{max+}/C_0 < 0)$, occurs at $T = -1.39$ or $1.39$ (i.e., at the phase of the inflection point in Figure 7a). Further, at a given $y/h_0$ and $-6.00 \leq T \leq -1.39$ or $1.39 \leq T \leq 6.00$, $v(T)/C_0$ increases from nearly zero to $[v(y/h_0)]_{max+}/C_0$ or decreasing from $[v(y/h_0)]_{max-}/C_0$ to almost zero. As a contrast, for $-1.39 \leq T \leq 1.39$, it does decrease from the maximum, $[v(y/h_0)]_{max+}/C_0$, via zero at $T = 0$, to $[v(y/h_0)]_{max-}/C_0$. It should be emphasized that, at the SMS and with increasing $y/h_0$ values, the "magnitudes" of vertical velocity remain increasing in the FSZ, in which the unsteady free-stream velocities are almost identical.

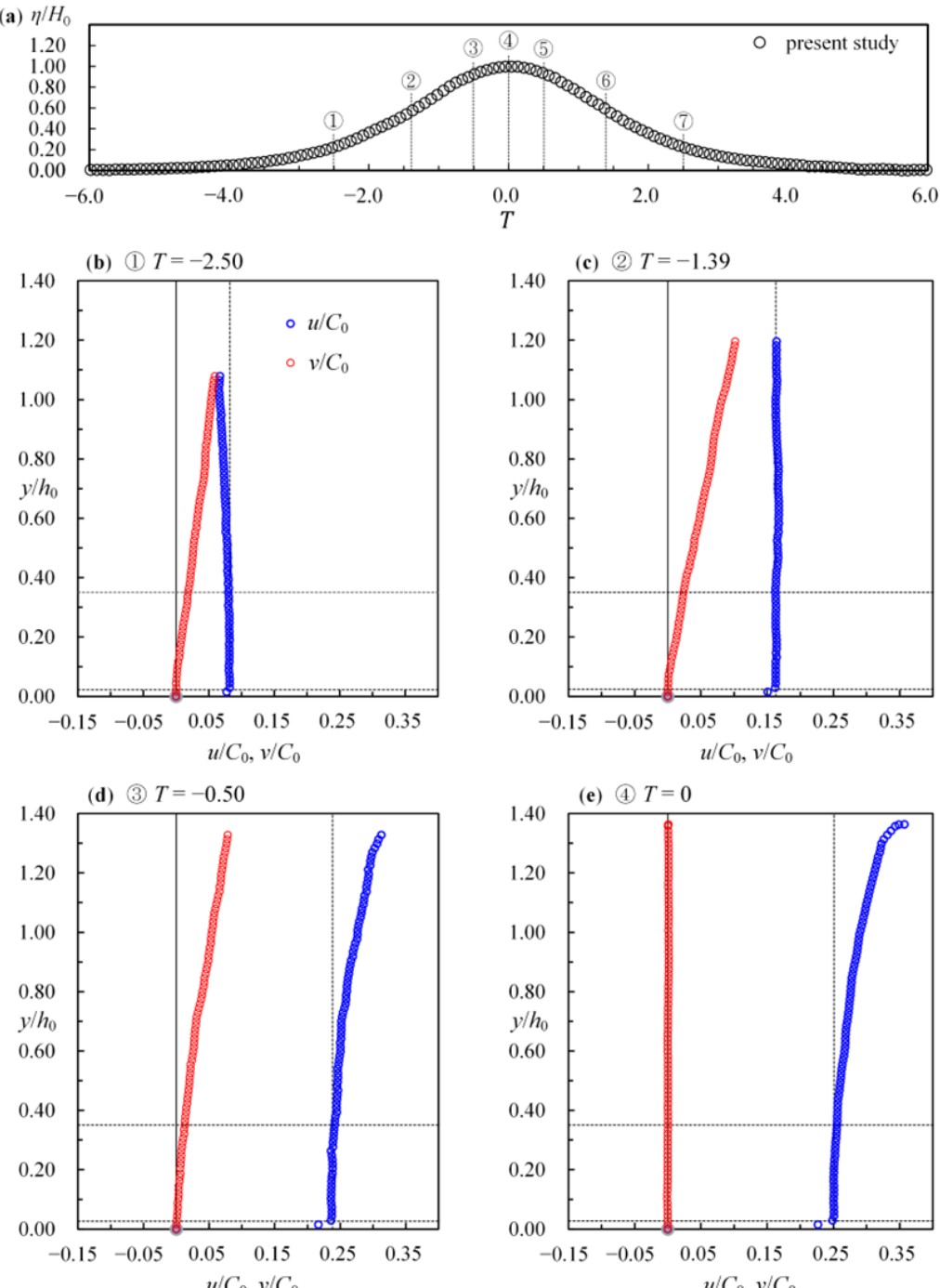

**Figure 7.** *Cont.*

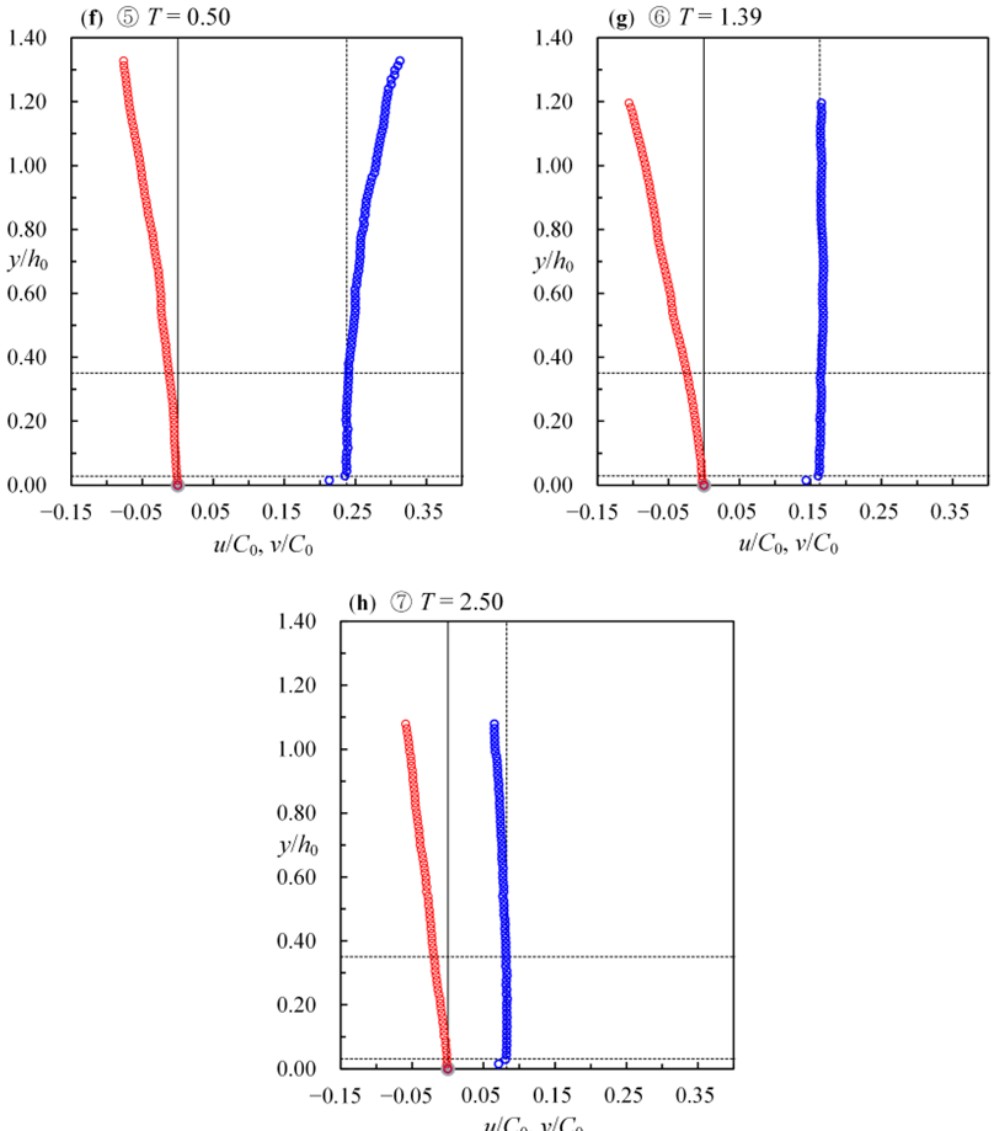

**Figure 7.** (**a**) Time series of dimensionless free surface elevation, $\eta(T)/H_0$; Dimensionless horizontal and vertical velocities profiles are shown at $T$ = (**b**) $-2.50$, (**c**) $-1.39$, (**d**) $-0.50$, (**e**) 0, (**f**) 0.50, (**g**) 1.39, and (**h**) 2.50 (Case E).

Figure 8a–g show close-ups for the temporal variation of horizontal and vertical velocity profiles which are obtained in both the boundary layer and the FSZ at $T = -2.50$, $-1.39$, $-0.50$, 0, 0.50, 1.39, and 2.50 for Case E. As similarly indicated in Figure 7b–h, the uniform free-stream velocities $u_o(T)$ exist in the FSZ situated inside the range within the two horizontal dotted lines. Further, the boundary layer is located inside the zone between the lower horizontal dotted line and the bed (i.e., $y/h_0 = 0$). The boundary layer thickness is defined herewith as a specified height measured upwards from the bed to the height where the horizontal velocity $u(T)$ at the edge of the boundary layer is equal to $0.99u_o(T)$, as shown at the lower horizontal dotted lines in Figure 8a–g. Note that, as primarily evidenced in Figure 8a to Figure 8g, the boundary layer thickness does increase slightly as $T$ varies from $-6.00$ to 6.00. More details relevant to the kinematic features and unique similarity profiles in the boundary layer flows can be referred to in Lin et al. [18].

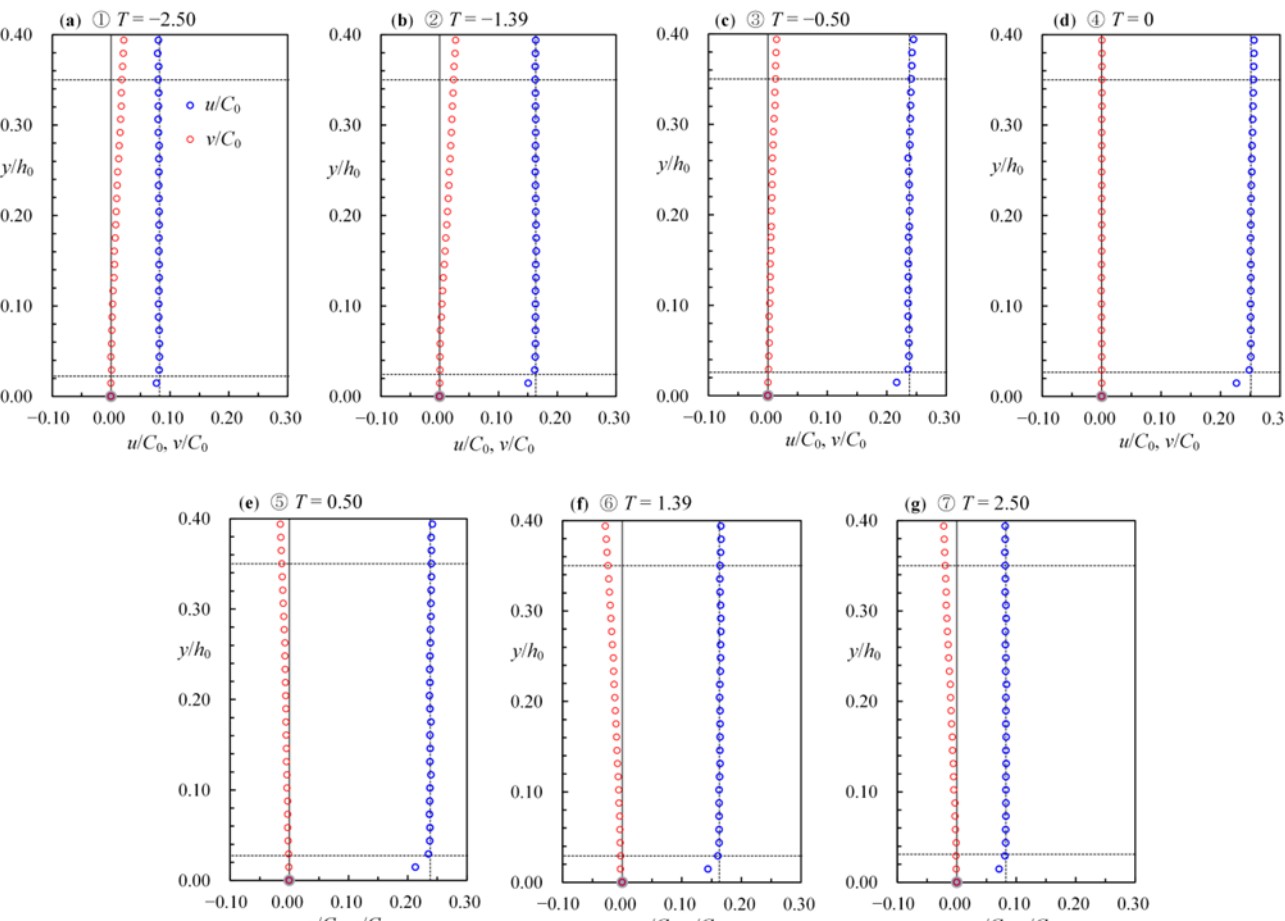

**Figure 8.** Temporal variation of the horizontal velocity profile obtained in the boundary layer and the FSZ at $T =$ (**a**) $-2.50$; (**b**) $-1.39$; (**c**) $-0.50$; (**d**) 0; (**e**) 0.50; (**f**) 1.39; and (**g**) 2.50 (Case E). Note that the FSZ is located within the region as marked by the two horizontal dashed lines, and that the internal zone or boundary layer is situated beyond the upper or beneath the lower horizontal dashed line.

For Case E at the SMS, Figure 9 presents the time series of dimensionless horizontal velocity $u(T)/C_0$ obtained at eight heights for $y/h_0 = 0.03$–$0.55$. For any one of the time series, horizontal velocity increases from near zero to a maximum of $-6.00 \leq T < 0$, indicative of the temporal acceleration in the horizontal direction. For $0 < T \leq 6.00$, $u(T)/C_0$ decreases from its maximum to near zero, suggestive of the temporal deceleration in the horizontal direction. The two features highlight the temporal acceleration equal to zero for $T = 0$, corresponding to which the maximum horizontal velocity occurs. For $y/h_0 = 0.05$–$0.35$ and $T = 0$, the maximum $u/C_0$ values are almost the same (about 0.251), smaller than those at $y/h_0 = 0.39$–$0.55$ in the internal zone and larger than those at $y/h_0 = 0.03$–$0.04$ in the bottom boundary layer. Further, the temporal variation in $u(T)/C_0$ at $y/h_0 = 0.05$ or 0.35 collapses completely onto those at $y/h_0 = 0.06$–$0.31$. These evidences strongly indicate that the FSZ exists for $y/h_0 = 0.05$–$0.35$, in which the magnitudes of free-stream velocities are nearly identical and the horizontal velocity profile is uniform. For the cases examined excluding Case E, the FSZs are located between $y/h_0 = (0.035{\sim}0.055)$ and $(0.335{\sim}0.366)$, which are almost equivalent to $y/h_0 = 0.05$–$0.350$ for Case E. For ease of data analysis, the flow velocities that represent kinematic features of FSZs are mined from the velocity fields only between $y/h_0 = 0.06$ and $0.330$.

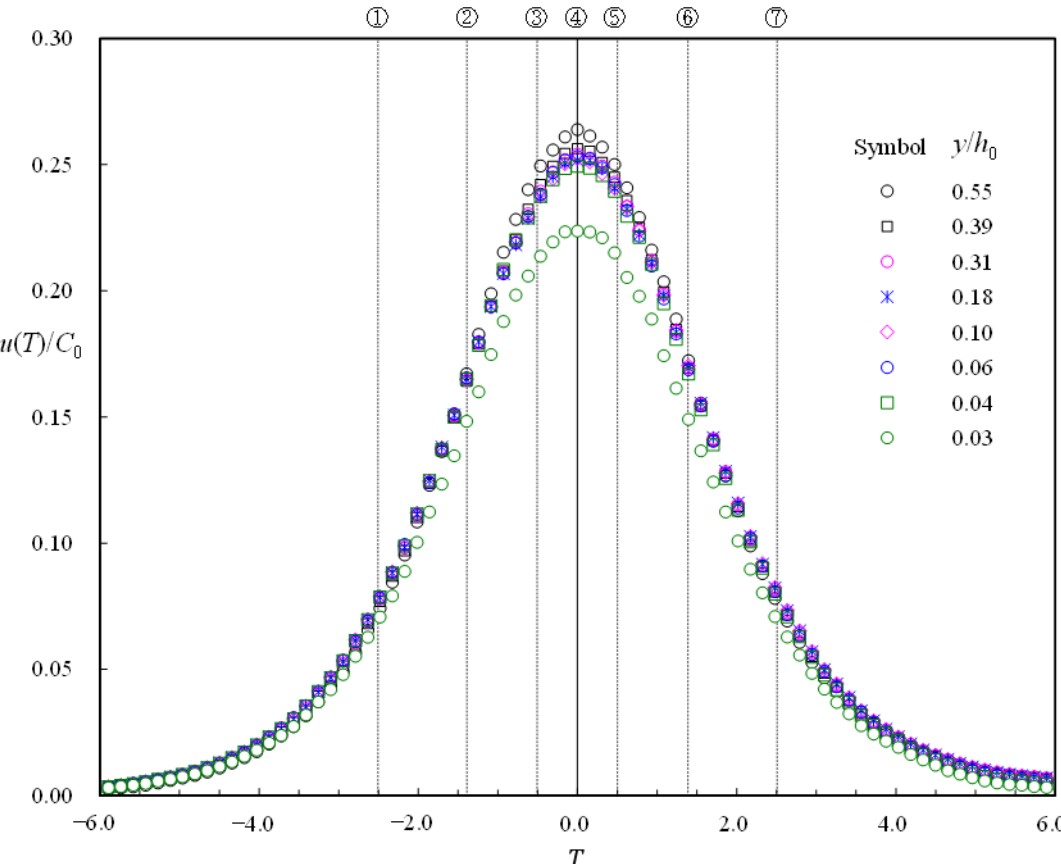

**Figure 9.** Temporal variation of $u(T)/C_0$ for $y/h_0 = 0.03$–$0.55$ (Case E).

In the following, the data of the nine cases ($H_0/h_0 = 0.050$–$0.550$, Table 1) are analyzed to explore the effect of nonlinearity (or $H_0/h_0$) not only on the dimensionless free surface elevations; but also on the dimensionless free-stream velocities, horizontal accelerations and pressure gradients in the FSZs of solitary waves.

*4.2. Nonlinear Effect on Free Surface Elevation and Free-Stream Velocity*

Figure 10 shows a systematic comparison of the relationships between $\eta(T)/H_0$ and $T$ for Cases A–I ($H_0/h_0 = 0.050$–$0.550$), along with a comparison of the measured data with the solitary wave theory [31] for each case. It is found that the experimental data are in good agreement with the analytic results for all cases. With an increase in $H_0/h_0$, the dimensionless free surface elevation becomes more focused around $T = 0$ with a narrower bell shape, exhibiting the $H_0/h_0$ (or nonlinearity) effect on the free surface profile. In other words, for a solitary wave with a larger $H_0/h_0$, a shorter time is taken to generate a complete wave motion. For example, for $\eta(T)/H_0 = 5\%$, the corresponding dimensionless times are examined to be $|T| = 11.47$, $5.85$, $4.12$, and $2.85$ for Case A ($H_0/h_0 = 0.050$), Case C ($H_0/h_0 = 0.179$), Case E ($H_0/h_0 = 0.363$), and Case I ($H_0/h_0 = 0.550$). The fact demonstrates the larger the $H_0/h_0$, the narrower the temporal range of wave shape. From the physical point of view, this trend shows that the change of ascending or descending free surface elevation per unit time becomes greater in magnitude. Namely, the free-surface slope $\partial[\eta(T)/H_0]/\partial T$ gets larger at a specified $T$ at a greater $H_0/h_0$, except those at $\eta(T) = 0$ and $\eta(0) = H_0$. It is noted that, as shown in Figure 11 for $-14.0 \leq T \leq 14.0$, the averaged free surface elevation $[\eta(T)/H_0]_{\text{mean}}$ decreases with an increase in $H_0/h_0$, rendering the relationship expressed by

$$[\eta(T)/H_0]_{\text{mean}} = 0.0737(H_0/h_0)^{-0.545}. \tag{1}$$

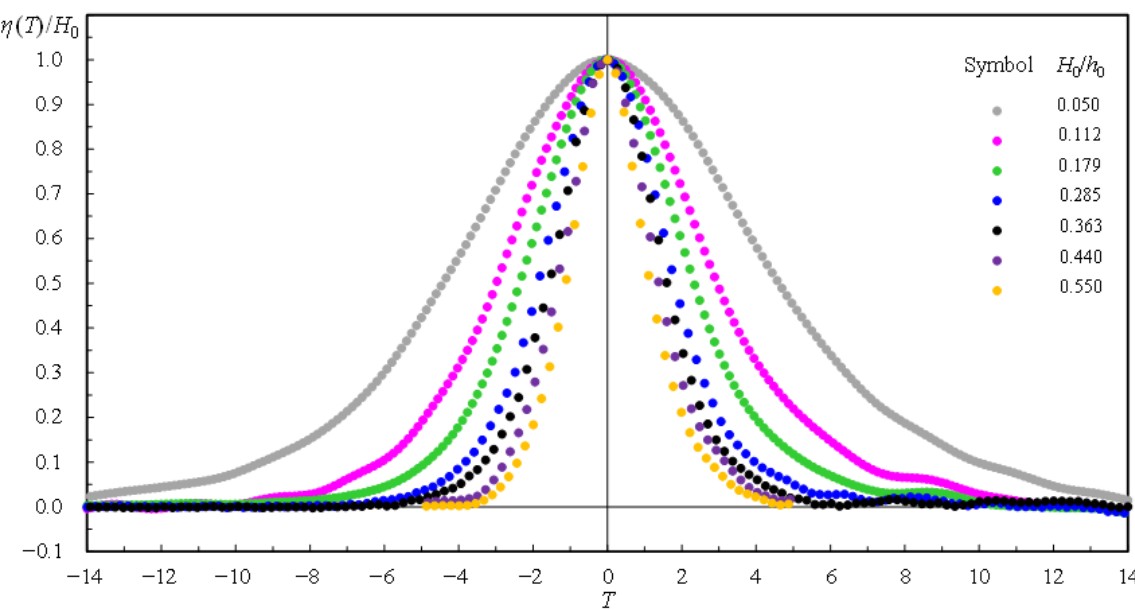

**Figure 10.** A comparison of temporal variations in the non-dimensional free surface elevation for $H_0/h_0 = 0.050$–$0.550$.

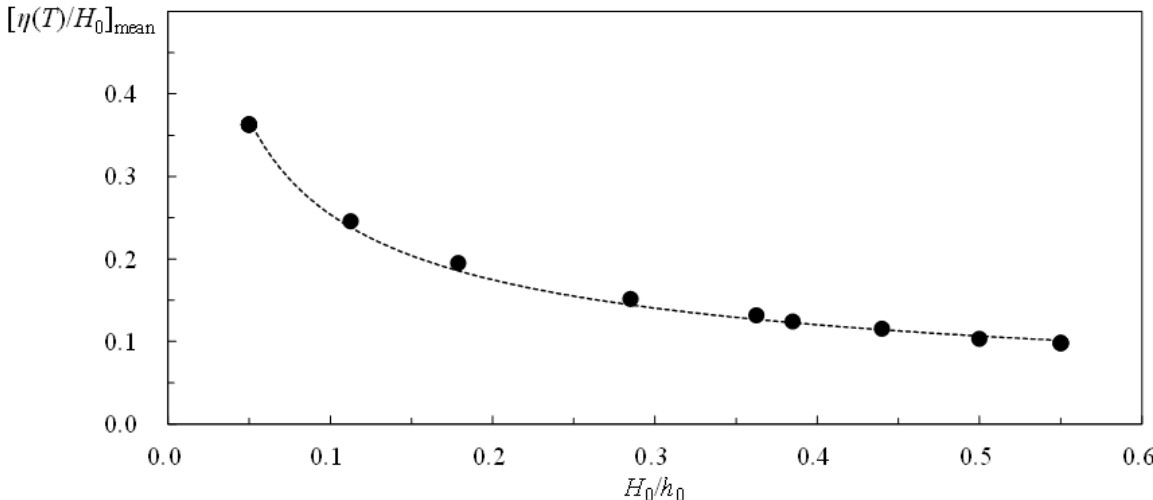

**Figure 11.** The relationship of $[\eta(T)/H_0]_{mean}$ versus $H_0/h_0$ for $-14.0 \leq T \leq 14.0$.

The result is attributable to $[\eta(T)/H_0]_{max} = 1.0$ for all experimental cases and the larger value of $H_0$ involved in the greater magnitude (i.e., higher nonlinearity) of $H_0/h_0$ at the same $h_0$ (= 8.0 cm).

Figure 12 presents a thorough comparison of the temporal variations of dimensionless free-stream velocities in the FSZs, $u_o(T)/C_0$, for Cases A–I ($H_0/h_0 = 0.050$–$0.550$). The nonlinear wave celerity $C_0$ (= $[g(H_0 + h_0)]^{1/2}$) is used herewith as the velocity scale. It is found that, at a larger $H_0/h_0$, the time series of $u_o(T)/C_0$ becomes more concentrated around $T = 0$ with a sharper bell shape. Less time is thus taken to achieve a complete variation of the free-stream velocities in the FSZ. The fact exhibits the $H_0/h_0$ (or nonlinearity) effect on the temporal distribution of free-stream velocity. Taking $u_o(T)/C_0 = 0.5\%$ for instance, the dimensionless times are identified to be about $|T| = 9.56$, 6.77, 5.58, and 4.65 for Case A ($H_0/h_0 = 0.050$), Case C ($H_0/h_0 = 0.179$), Case E ($H_0/h_0 = 0.363$), and Case I ($H_0/h_0 = 0.550$), respectively. These data do testify to the relatively narrower temporal range of free-stream velocity induced by a solitary wave with a greater value of $H_0/h_0$. At a specified dimensionless time, say $|T| < 3.95$, this trend does show that the change rate of

free-stream velocity, $\partial[u_o(T)/C_0]/\partial T$ (equivalent to the dimensionless local acceleration in the FSZ), becomes larger in magnitude. For $T < 0$ or $T > 0$, $\partial[u_o(T)/C_0]/\partial T$ is positive or negative, indicative of flow acceleration or deceleration in the FSZ for each case.

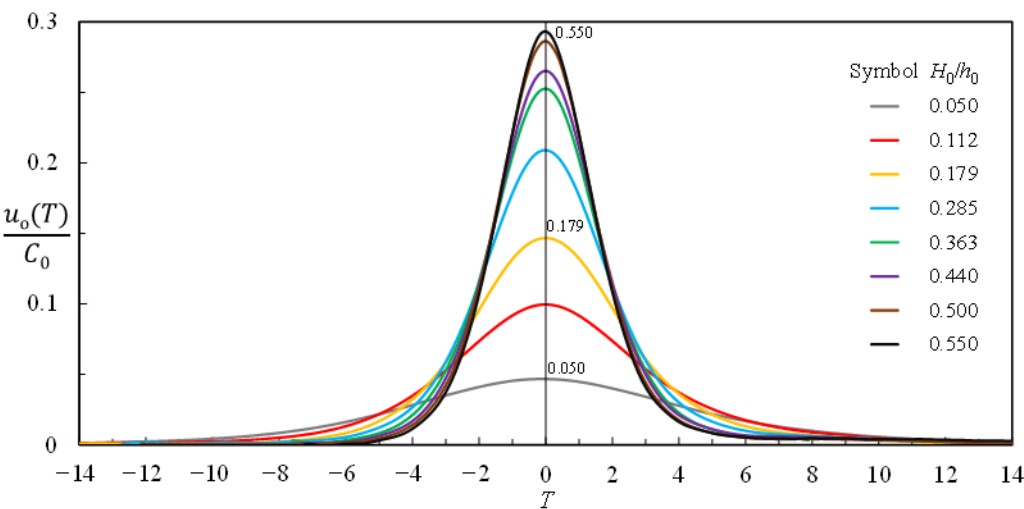

**Figure 12.** A comparison of temporal variations in the dimensionless free-stream velocity for $H_0/h_0 = 0.050$–$0.550$.

In addition, the maximum dimensionless free-stream velocity, $[u_o/C_0]_{max}$, occurs with zero acceleration at $T = 0$. As evidenced in Figure 13, $[u_o/C_0]_{max}$ becomes larger with an increasing $H_0/h_0$. Its overall nonlinear form is regressed with an $R^2$ value of 0.992 as:

$$[u_o/C_0]_{max} = -0.8208(H_0/h_0)^2 + 0.9789(H_0/h_0) \tag{2}$$

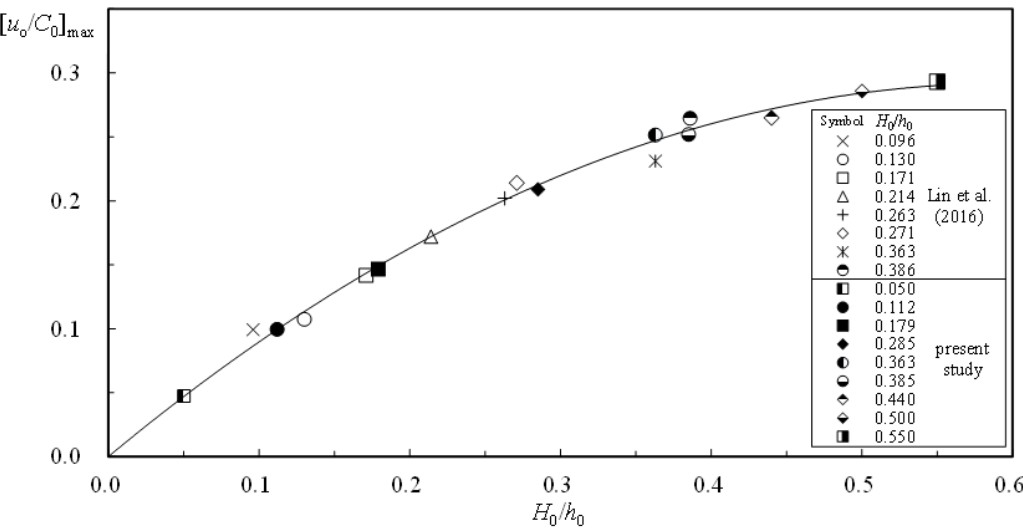

**Figure 13.** Relationship between the dimensionless maximum free-stream velocity and $H_0/h_0$.

For Cases C ($H_0/h_0 = 0.179$), E ($H_0/h_0 = 0.363$), and I ($H_0/h_0 = 0.550$), the values of $[u_o/C_0]_{max}$ are about 3.10, 5.32, and 6.20 times that (= 0.0473) for Case A ($H_0/h_0 = 0.050$), clearly substantiating the nonlinear effect on $[u_o/C_0]_{max}$. Further, as illustrated in Figure 14 for $-14.0 \leq T \leq 14.0$, the averaged free-stream velocity $[u_o(T)/C_0]_{mean}$ increases if $H_0/h_0$ increases. The relationship between the former and the latter is written as

$$[u_o(T)/C_0]_{mean} = -0.2962(H_0/h_0)^4 + 0.6725(H_0/h_0)^3 - 0.5646(H_0/h_0)^2 + 0.2123(H_0/h_0) + 0.0088 \tag{3}$$

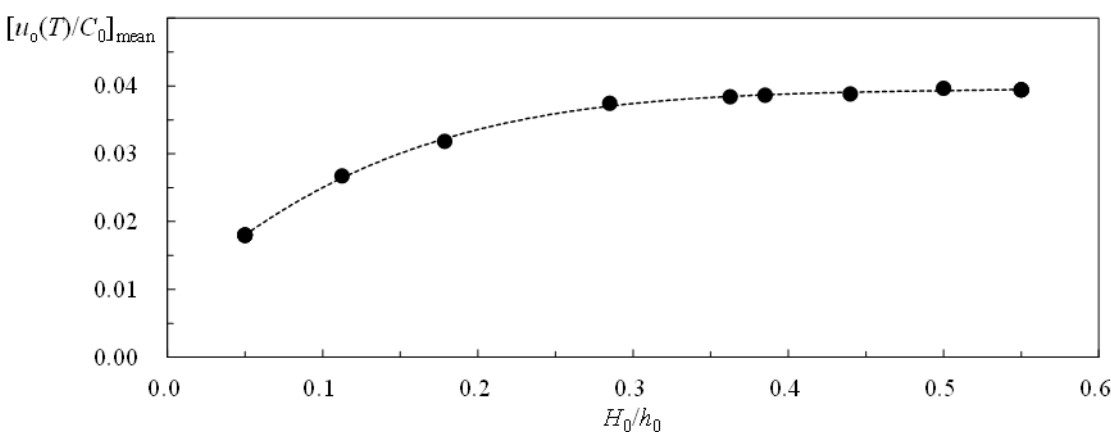

**Figure 14.** The relationship between $[u_o(T)/C_0]_{mean}$ and $H_0/h_0$ for $-14.0 \leq T \leq 14.0$.

*4.3. Nonlinear Effect on Local and Convective Accelerations in FSZ*

For a solitary wave traveling over a horizontal bed, the local and convective accelerations in the horizontal direction [33,34] are defined as:

$$A_l(x, y, t) = \partial u(x, y, t)/\partial t \tag{4}$$

And

$$A_c(x, y, t) = u(x, y, t) \times \partial u(x, y, t)/\partial x + v(x, y, t) \times \partial u(x, y, t)/\partial y \tag{5}$$

Within the FSZ, the unsteady free-stream velocities $u_o(x, y, t)$ $[= u_o(x, t)]$ are uniform in the vertical direction, thus leading to $\partial u_o(x, t)/\partial y = 0$. Therefore, at the SMS and in the FSZ, the local and convective accelerations in the horizontal direction are simplified as:

$$A_{lo}(0, t) = \partial u_o(0, t)/\partial t \approx \Delta u_o(0, t)/\Delta t = A_{lo}(t) \tag{6}$$

And

$$A_{co}(0, t) = u_o(0, t) \times \partial u_o(0, t)/\partial x \approx u_o(0, t) \times \Delta u_o(0, t)/\Delta x = A_{co}(t). \tag{7}$$

Accordingly, the values of $A_{lo}(t)$ and $A_{co}(t)$ can be computed, using a central difference scheme, from the instantaneous free-stream velocities obtained at the SMS and in its close neighborhood. From a theoretical point of view, the time and spatial interval, $\Delta t$ and $\Delta x$, should be as small as possible (i.e., approaching zero). However, as reported by Lin et al. [19,20,25,27], even the use of the smallest time or spatial interval in the differential computation would result in failure to find a convergent $A_{lo}(t)$ or $A_{co}(t)$ value. The reason is attributable to the rapid temporal or spatial fluctuations in the "image-based (or pixel-based)" PIV/HSPIV measurements.

Following the method proposed by Lin et al. [25,27] a range of the promising time or spatial intervals, in which the relative deviation of each computed outcome of $A_{lo}(t)$ or $A_{co}(t)$ would only vary less than 4.0% of its average, is selected. Detailed illustrations for the trial-and-error calculation of $A_{lo}(t)$ and $A_{co}(t)$ with different $\Delta t$ or $\Delta x$ values, together with the determination of promising time or spatial interval, can be referred to Lin et al. [25,27]. In this study, the promising time or interval used is equal to $(\Delta t)_{promising}$ = 0.01–0.0156 s, (i.e., 25 times $(\Delta t)_{framing}$ [= 1/2500–1/1600 s = 0.0004–0.000625 s]) or $(\Delta x)_{promising}$ = 0.10712–0.248 cm (i.e., 13–31 times the grid size $(\Delta x)_{gs}$ [= $(\Delta y)_{gs}$ = 0.00824 or 0.008 cm] used in the HSPIV measurements). It should be emphasized that previous investigations [25,27,35] did indicate $(\Delta t)_{promising}$ or $(\Delta x)_{promising}$ much larger than $\Delta t_{framing}$ or $(\Delta x)_{gs}$, but not as small as $(\Delta t)_{framing}$ or $(\Delta x)_{gs}$. Like the data processing used in Jensen et al. [35], a symmetric 7-point

smoothing scheme with distinct weightings was utilized to smoothen the time series of $A_{lo}(t)$ or $A_{co}(t)$.

At the SMS, Figure 15 displays a complete comparison of the temporal variations of dimensionless local acceleration in the FSZs, $A_{lo}/g$, for Cases A–I ($H_0/h_0$ = 0.050–0.550). Note that $A_{lo}(T)/g \approx -A_{lo}(-T)/g$, exhibits *the odd-function feature*. As addressed in Figure 8, the free-stream velocity increases from near zero to a maximum for $-6.00 \leq T < 0$ or decreases from the maximum to about zero for $0 < T \leq 6.00$, underscoring the temporal acceleration or deceleration in the FSZ. Accordingly, for all cases at the SMS, $A_{lo}/g$ is positive for $-6.00 \leq T < 0$ and negative for $0 < T \leq 6.00$. Note that $A_{lo}/g = 0$ for $T = 0$, at which the wave crest moves right over the SMS and the free-stream velocity arrives at its maximum. It is interesting to mention that the averaged value of the dimensionless local acceleration, $[A_{lo}(T)/g]_{mean}$, is equal to almost zero for $-6.00 \leq T \leq 6.00$, which is attributable to the time series of $A_{lo}(T)/g$ featured with an odd-function form.

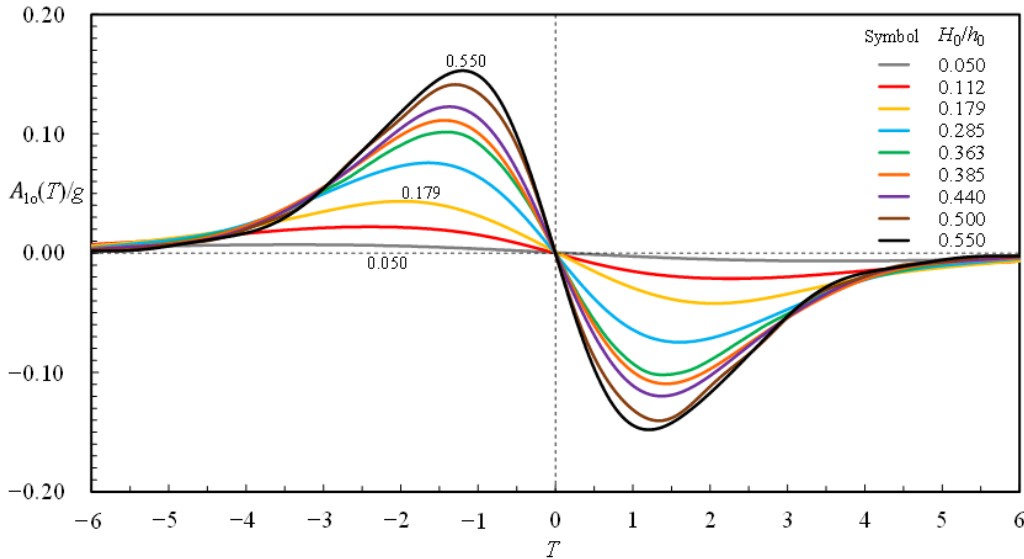

**Figure 15.** A comparison of temporal variations in the non-dimensional local acceleration for $H_0/h_0$ = 0.050–0.550.

As also seen in Figure 15, the positive and negative maxima of $A_{lo}/g$, $A_{lo+}/g$, and $A_{lo}-/g$ ($\approx -A_{lo+}/g < 0$), highlight almost equal magnitude at the dimensionless characteristic time $T = T_{Alo+}$ ($< 0$) and $T = T_{Alo-}$ ($> 0$), respectively. As illustrated in Figures 7 and 8, the translation of free-stream velocity is in phase with the free-surface motion of a solitary wave. Therefore, $T_{Alo+}$ and $T_{Alo-}$ correspond, in fact, to the inflection points in the time series of both $\eta(T)/H_0$ and $u_o(T)/C_0$. Namely, the slopes of $\eta(T)/H_0$ and $u_o(T)/C_0$ (i.e., $\partial[\eta(T)/H_0]/\partial T$ and $\partial[u_o(T)/C_0]/\partial T$) exhibit the local maxima apparently at $T_{Alo+}$ and $T_{Alo-}$. As evidenced in Figure 16, $A_{lo+}/g$ and $A_{lo}-/g$ increase linearly, and $T_{Alo+}$ and $T_{Alo-}$ decrease with an increasing $H_0/h_0$ (or nonlinearity). For example, $A_{lo+}/g$ ($= -A_{lo}-/g = 0.0434, 0.1075$, and $0.1523$) for Cases C, E and I are about 6.11, 15.14, and 21.45 times that ($= 0.0071$) for Case A, demonstrating the obvious nonlinear effect on $A_{lo+}/g$ and $A_{lo}-/g$. The reason is assignable to $\eta(T)/H_0$ and $u_o(T)/C_0$ both becoming more concentrated around $T = 0$ with narrower symmetric bell shapes for a greater $H_0/h_0$. Namely, if $H_0/h_0$ gets larger, a shorter time is taken to achieve a complete motion with a greater value of the maximum free-stream velocity (i.e., translating with relatively larger $|\Delta u|$ under the same $\Delta t$), thus resulting in a larger $\partial[u_o(T)/C_0]/\partial T$.

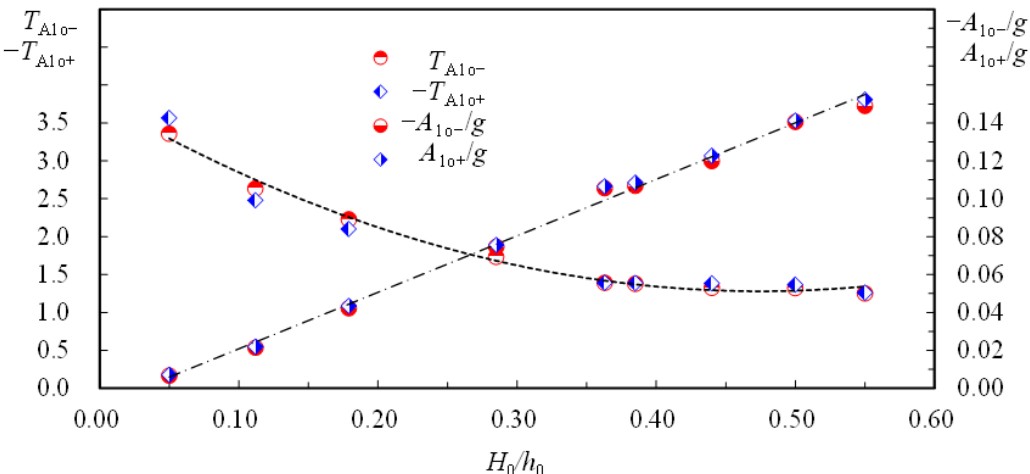

**Figure 16.** The relationships of $T_{Alo+}$ and $T_{Alo-}$ as well as $A_{lo+}/g$ and $A_{lo-}/g$ versus $H_0/h_0$.

Figure 17 shows a systematic comparison of the time series of dimensionless convective acceleration in the FSZs, $A_{co}/g$, for Cases A–I ($H_0/h_0$ = 0.050–0.550). As illustrated in Figure 3, $t < 0$, $t = 0$, and $t > 0$ correspond to the times before, at, and after the wave crest intersection with the SMS. This leads to the maximum free-stream velocity, $[u_o(x)]_{max}$, appearing at $x < 0$, $x = 0$ or $x > 0$; and subsequently, $\partial u_o(t)/\partial x < 0$, $\partial u_o(0)/\partial x = 0$ or $\partial u_o(t)/\partial x > 0$ at the SMS. Accordingly, $A_{co}/g$ ($= [u_o(t) \times \partial u_o(0, t)/\partial x]/g$) takes negative, zero, or positive values for $T < 0$, $T = 0$, or $T > 0$, as observed in Figure 17 almost with *the odd-function feature* [i.e., $A_{co}(T)/g \approx -A_{co}(-T)/g$]. For all cases, the magnitudes of negative and positive maxima in the dimensionless convective acceleration, $A_{co-}/g$ and $A_{co+}/g$, occur at the dimensionless characteristic time $T = T_{Aco-}$ ($< 0$) and $T = T_{Aco+}$ ($> 0$), and increase with increasing $H_0/h_0$. Note that $|T_{Aco-}|$ and $T_{Aco+}$ are smaller than $|T_{Alo+}|$ and $T_{Alo-}$. For example, for Case E, $-T_{Aco-} = T_{Aco+} \approx 1.090 < 1.390 \approx -T_{Alo+} = T_{Alo-}$. The fact points out that the dimensionless times for the occurrence of $A_{co-}/g$ and $A_{co+}/g$ are closer to $T = 0$ than the counterparts of $A_{lo+}/g$ and $A_{lo-}/g$. Further, $A_{co-}/g$ and $A_{co+}/g$ are about 1/22.0–1/4.3 times $A_{lo+}/g$ and $A_{lo-}/g$, justifying much more contribution from the local acceleration than from the convection acceleration in the pressure gradient. Further, the averaged value of the dimensionless convective acceleration, $[A_{co}(T)/g]_{mean}$, is equal to almost zero for $-6.00 \leq T \leq 6.00$ because the time series of $A_{co}(T)/g$ is characterized by an odd-function form.

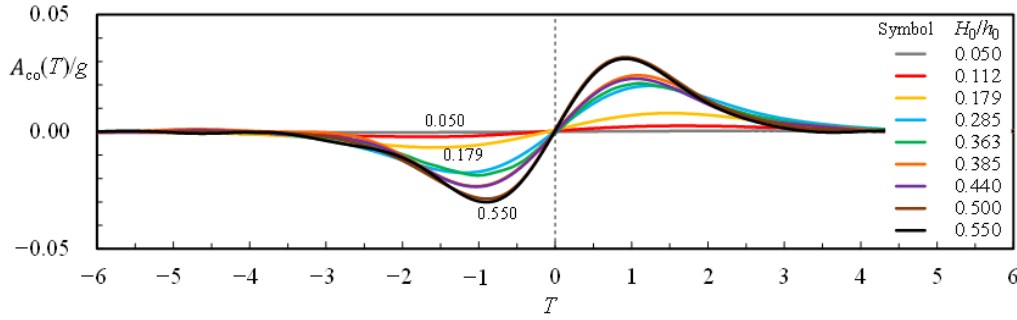

**Figure 17.** A comparison of temporal variations in the non-dimensional convective acceleration for $H_0/h_0$ = 0.050–0.550. Note that the vertical scale used herein is distinct from that shown in Figure 15.

*4.4. Nonlinear Effect on Pressure Gradient in FSZ*

Based on the incompressible Navier-Stokes equations [33,34] for a solitary wave traveling over a horizontal bed, the gravity term is equal to zero and the contribution of the

viscous term is negligibly small [19,20,25] and thus omitted. Accordingly, the horizontal pressure gradient in the FSZ at the SMS is expressed as:

$$(1/\rho)\partial p_o(0, t)/\partial x = -[\partial u_o(0, t)/\partial t + u_o(0, t) \times \partial u_o(0, t)/\partial x]$$
$$= -[A_{lo}(t) + A_{co}(t)]. \tag{8}$$
$$= P_o(t)$$

Thus, $P_o(t)$ takes a minus value of $[A_{lo}(t) + A_{co}(t)]$ (i.e., *the force per unit mass in the horizontal direction*). With the magnitude of $A_{lo}(t)$ being much larger than $A_{co}(t)$, $A_{lo}(t)$ dominates the contribution to $P_o(t)$.

Figure 18 shows a thorough comparison of the temporal variations in dimensionless pressure gradient $P_o(T)/g$ for all cases. The characteristic dimensionless times for the occurrence of a negative maximum $[P_o(T)/g]_{max-}$ $(= P_{o-}/g)$ and a positive maximum $[P_o(T)/g]_{max+}$ $(= P_{o+}/g)$ are defined as $T_{Po-}$ $(< 0)$ and $T_{Po+}$ $(> 0)$, respectively. Similar to the variation feature of $A_{lo}(T)/g$ (see Figure 15), an *odd-function form* of $P_o(T)/g$ is examined by $P_{o+}/g \approx -P_{o-}/g$ and $T_{Po+} = T_{Alo-} \approx -T_{Alo+} = -T_{Po-}$. $P_o(T)/g$ decreases from near zero to $P_{o-}/g$ for $-6.00 \leq T \leq T_{Po-}$ or from $P_{o+}/g$ to about zero for $T_{Po+} \leq T \leq 6.00$, indicating an increase in the favorable pressure gradient or a decrease in the adverse pressure gradient in the FSZ. Further, it increases from $P_{o-}/g$ $(< 0)$, via 0, to $P_{o+}/g$ $(> 0)$ for $T_{Po-} \leq T < 0, T = 0$, and $0 < T \leq T_{Po+}$, respectively, demonstrating that $P_o(T)/g$ varies correspondingly from favorable, via zero, to the adverse pressure gradient. Note that, for $-6.00 \leq T \leq 6.00$, the averaged value of the dimensionless pressure gradient, $[P_o(T)/g]_{mean}$, is equal to almost zero because the time series of $P_o(T)/g$ is featured with an odd-function form.

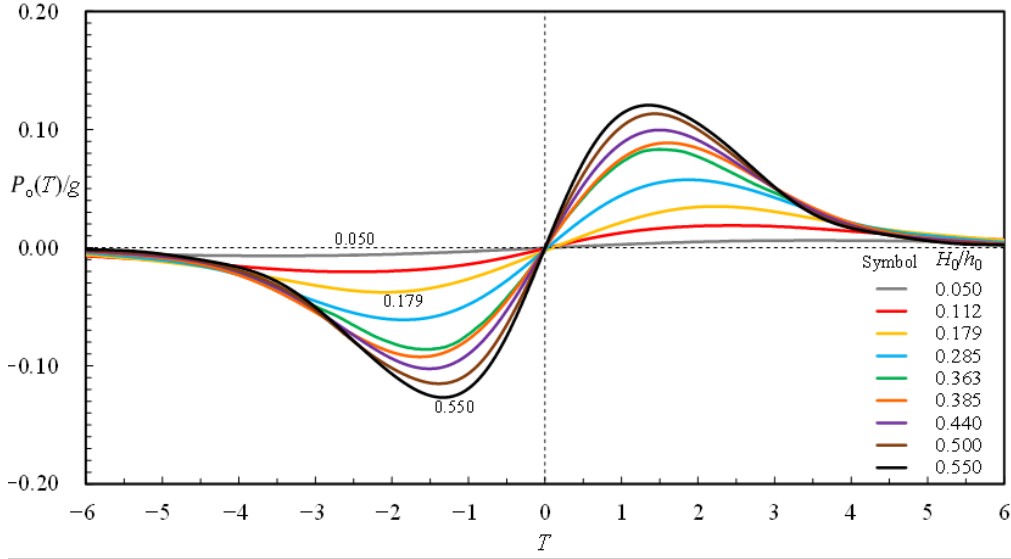

**Figure 18.** A comparison of temporal variations in the non-dimensional pressure gradient of the free stream for $H_0/h_0 = 0.050$–$0.550$.

Figure 19 outlines the relationships of $P_{o+}/g$ $(\approx -P_{o-}/g)$ and $T_{Po+}$ $(\approx -T_{Po-})$ versus $H_0/h_0$. It is found that $P_{o+}/g$ increases and $T_{Po+}$, however, decreases with an increasing $H_0/h_0$. Namely, at a larger $H_0/h_0$ value, $P_{o+}/g$ becomes greater and takes place closer to $T = 0$ (i.e., the "phase" with wave crest right passing through the SMS). For example, for Cases C, E, and I, $P_{o+}/g = 0.0350, 0.0887$, and $0.1210$ appear at $T_{Po+} = 0.1674, 0.1049$, and $0.0943$, respectively. The values of $P_{o+}/g$ are 5.74, 14.54, and 19.84 times that of Case A $(= 0.0061$, with $T_{Po+} = 0.2463)$, underlining the distinct effect of nonlinearity on $P_o(T)/g$ and $P_{o+}/g$.

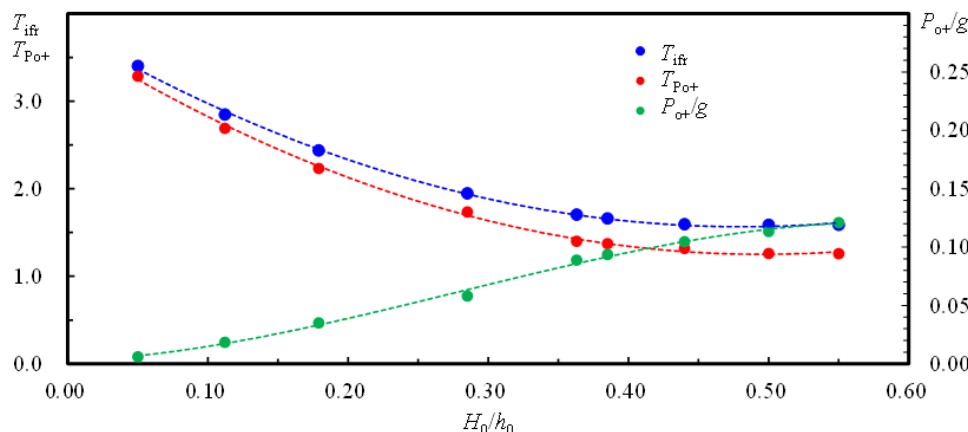

**Figure 19.** Variation trends of the non-dimensional times for occurrence of flow reversal and maximum adverse pressure gradient, and of the values of maximum adverse pressure gradient for $H_0/h_0$ = 0.050–0.550.

Here an interesting issue arises relevant to whether the occurrence of incipient flow reversal precedent to, simultaneous with, or later than the appearance of a maximum adverse pressure gradient. Namely, to examine $T_{\text{Po+}} > T_{\text{ifr}}$, $T_{\text{Po+}} = T_{\text{ifr}}$, or $T_{\text{Po+}} < T_{\text{ifr}}$. In this study, $T_{\text{ifr}}$ is determined from the flow-visualized images. To well observe the incipient flow reversals with "particle-dotted" and "path-lined" images for all cases, the ranges of framing rate of the camera were set at 1600–2500 for the HSPIV measurements and 30–100 Hz for flow visualization tests. The former enhanced the temporal resolution (i.e., 1/2500–1/1600 s) in precisely identifying $T_{\text{ifr}}$ from the continuous display (with frame-by-frame operation) of the recorded images. The latter provided visual evidence related to the temporal variation of flow structure in the near-bottom zone. Together with a concise description of other cases, the observation results are here detailed only for Case E. For ease of understanding, an original motion picture displaying Case E's flow structure in Supplementary Material, in which eight images of the instantaneous flow structure are drawn and then elucidated as follows.

At the SMS and for $0 \leq T \leq 6.09$, Figure 20a–h presents a series of the instantaneous path-lined images of the flow field (left panels) and the schematic diagrams of the velocity profile (right panels) within $-0.10 \leq x/h_0 \leq 0.10$ and $0 \leq y/h_0 \leq 0.10$ for Case E. The water particles in the FSZ all move from the left to right side in each image, as evidenced by the temporal variation of free-stream velocities [i.e., $u_o(T) > 0$] at $y/h_0$ = 0.05–0.35 (Figure 8). The longer the length of each pathline, the larger the magnitude of each particle velocity. As observed in Figure 20a, the free-stream velocity reaches its maximum and $P_o/g = 0$ at $T = 0$. Immediately after the passing of wave crest through the SMS, say $0 < T \leq 1.67$, the lengths and the corresponding free-stream velocities in the FSZ decrease with an increasing $T$ (see Figure 20b,c for $T$ = 0.89 and 1.76). It is thus indicated that the flow in the near-bottom zone, subjected to adverse pressure gradients, does decelerate temporally.

Based on detailed observation from a continuous display (with forward and backward motions) of both path-lined and instantaneously particle-dotted images at the SMS, the local horizontal velocities are found to be almost zero for $y/h_0 < 0.004$ at $T = T_{\text{ifr}} = 1.67$ (Figure 20c). After $T > T_{\text{ifr}}$, flow reversal occurs right beyond the bed. Namely, the pathlines located very near the bed start translating from right to left, indicating that the particle velocities are negative, see Figure 20d–h at $T$ = 2.55–6.09. The counterparts observed in the FSZ move, however, from left to right. This reveals that the corresponding particle velocities are positive, but keep decreasing with $T$. The thickness of the flow reversal layer increases with an increasing $T$ if $T > T_{\text{ifr}}$. This feature shows that the particle velocities right on the bed and exactly at the edge of flow reversal are both zero with negative velocities in between. Interestingly note that, for Case E, the incipient flow reversal occurs at $T = T_{\text{ifr}} = 1.67$ (or $t = t_{\text{ifr}} = 0.1508$ s), soon after but not simultaneously with the occurrence

of $P_{o+}/g$ at $T = T_{Po+} = T_{Alo+} = 1.39$ (or $t = t_{Po+} = t_{Alo+} = 0.1255$ s). The reason for such a slight temporal delay with $(\Delta T)_{delay} = 0.28$ [or $(\Delta t)_{delay} = 0.0253$ s] is attributable to the viscous damping effect in the bottom boundary layer.

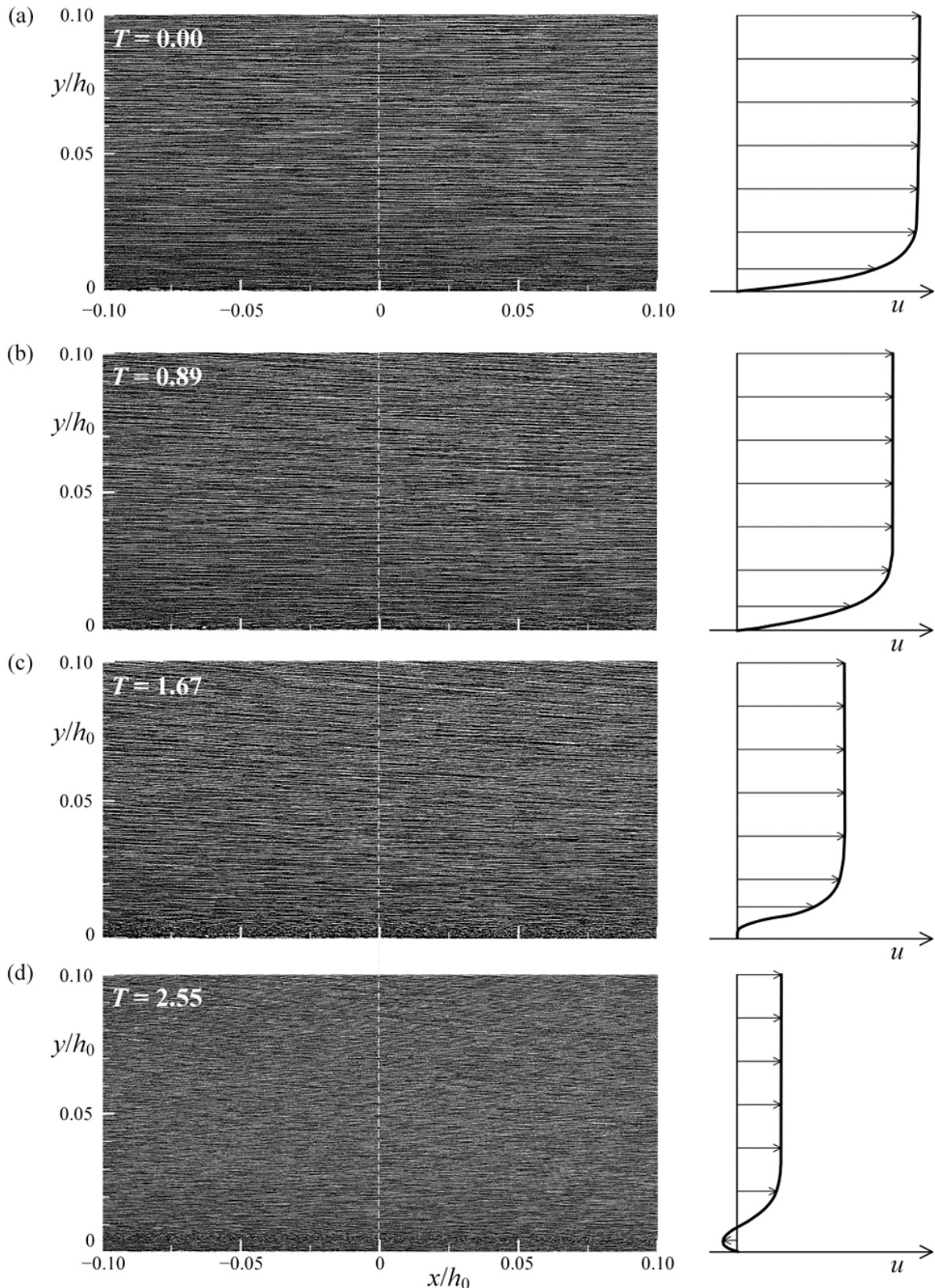

**Figure 20.** *Cont.*

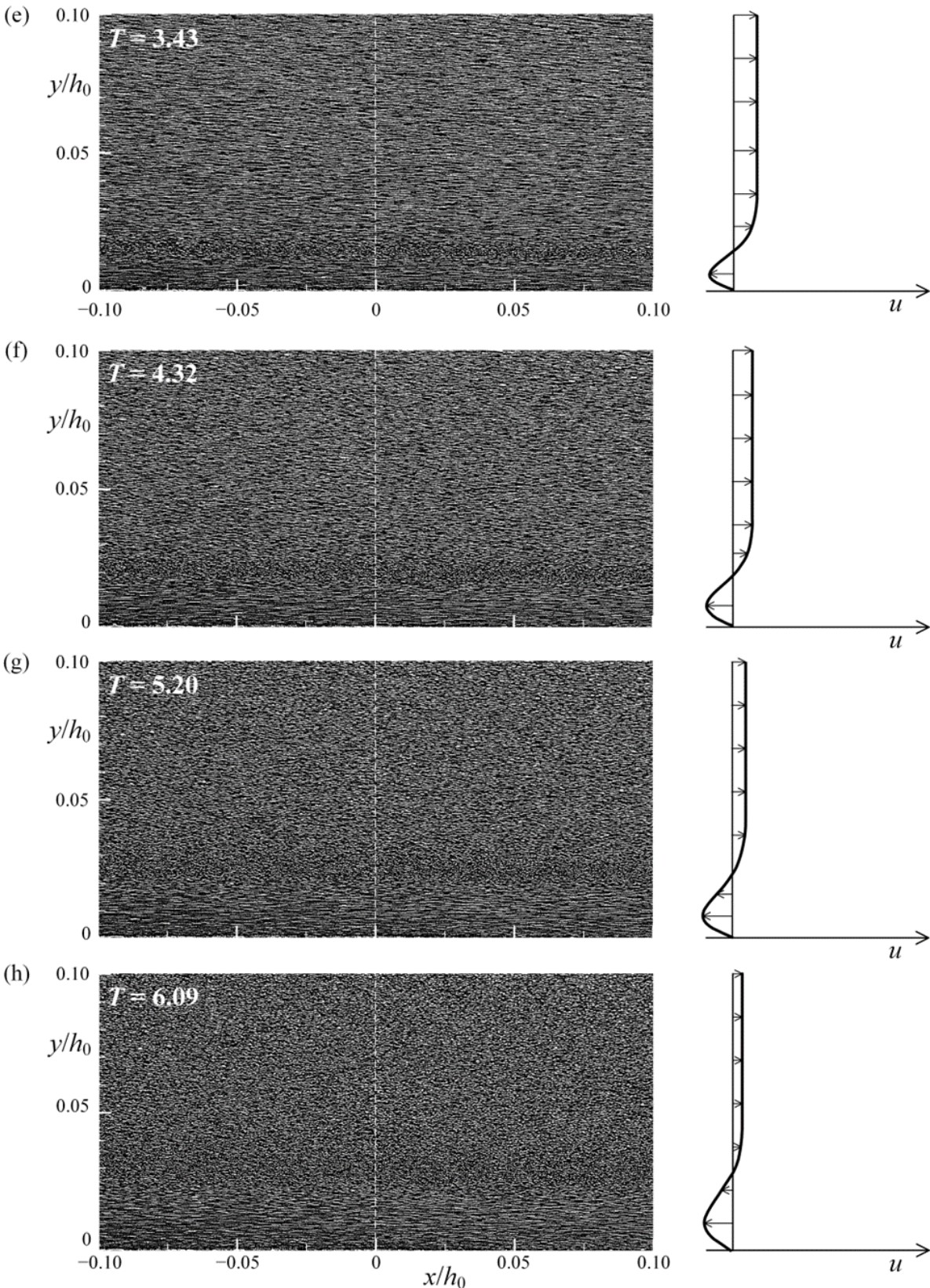

**Figure 20.** Flow-visualized images at $T =$ (**a**) 0; (**b**) 0.89; (**c**) 1.67; (**d**) 2.55; (**e**) 3.43; (**f**) 4.32; (**g**) 5.20; (**h**) 6.09 for $-0.10 \leq x/h_0 \leq 0.10$ and $0 \leq y/h_0 \leq 0.10$.

Till now, both the path-lined images (Figure 20a–h) and the temporal variations of $[P_o(T)]/g$ (Figure 18) have been demonstrated for the first time to shed light on the relationship between $T_{ifr}$ and $T_{Po+}$ (= $T_{Alo+}$). For each visualized case other than Case E, the incipient flow reversal does take place immediately after the maximum adverse pressure gradient.

In other words, $T_{ifr}$ is slightly larger than $T_{Po+}$. Finally, the relationship between $T_{ifr}$ and $H_0/h_0$ is also shown in Figure 19, indicating $T_{ifr}$ decreasing prominently with an increase in $H_0/h_0$. This demonstrates the effect of nonlinearity on the incipient flow reversal in the near-bottom zone.

## 5. Conclusions

In this study, with respect to the nonlinear effect on the kinematic and hydrodynamic features in the FSZs of solitary waves for $H_0/h_0$ = 0.050–0.550, a series of experimental results at the SMS are presented, systematically showing the results of the relevant flow parameters. The findings are summarized as follows.

1.  For all the cases exclusive of Case E, the FSZs are positioned between $y/h_0$ = (0.035–0.055) and (0.335–0.366), nearly identical to those between $y/h_0$ = 0.05 and 0.350 in Case E.
2.  If $H_0/h_0$ increases, the dimensionless free surface elevation, $\eta(T)/H_0$, and the dimensionless free stream velocity, $u_o(T)/C_0$, become more concentrated around $T = 0$ with a narrower symmetric bell-shape, exhibiting shorter time taken to generate a complete wave motion. This trend indicates that the change of ascending or descending free surface elevation per unit (dimensionless) time becomes greater in magnitude.
3.  For $-6.00 \leq T < 0$ and $0 < T \leq 6.00$, the dimensionless free-stream velocity, $[u_o(T)/C_0]$, increases from near zero to a maximum and decreases from the maximum to about zero, highlighting the temporal acceleration and deceleration in the FSZ.
4.  The relationship between $[u_o/C_0]_{max}$ and $H_0/h_0$ is uniquely expressed in Equation (3), stating that the former gets large with an increasing $H_0/h_0$. For $H_0/h_0$ = 0.179, 0.363 and 0.550, the values of $[u_o/C_0]_{max}$ are about 3.10, 5.32, and 6.20 times that (= 0.0473) for $H_0/h_0$ = 0.050. This trend demonstrates the nonlinear effect on $[u_o/C_0]_{max}$.
5.  The dimensionless local acceleration, $A_{lo}/g$, is positive for $-6.00 \leq T < 0$ and negative for $0 < T \leq 6.00$. At $T = 0$ with wave crest intersecting the SMS, $A_{lo}/g$ is equal to zero and the free-stream velocity reaches its maximum.
6.  The magnitudes of positive and negative maxima in the dimensionless local acceleration, $A_{lo+}/g$ and $A_{lo-}/g$ ($\approx -A_{lo+}/g$), increase linearly, and the counterparts of the dimensionless characteristic time $T_{Alo+}$ (< 0) and $T_{Alo-}$ (> 0) decrease with an increase in $H_0/h_0$. For $H_0/h_0$ = 0.179, 0.363 and 0.550, the values of $A_{lo+}/g$ are about 6.11, 15.14 and 21.45 times that (= 0.0071) for $H_0/h_0$ = 0.050, indicating the nonlinear effect on $A_{lo+}/g$ and $A_{lo-}/g$.
7.  The magnitudes of negative and positive maxima in the dimensionless convective acceleration, $A_{co-}/g$, and $A_{co+}/g$, increase when $H_0/h_0$ increases. However, their magnitudes are about 1/22.0–1/4.3 times those of $A_{lo+}/g$ and $A_{lo-}/g$. With the magnitude of $A_{co}(T)/g$ being much smaller than that of $A_{lo}(T)/g$, the contribution to the dimensionless pressure gradient, $P_o(T)/g$ (= $-[A_{lo}(T) + A_{co}(T)]/g$), is thus governed mainly by $A_{lo}(T)/g$.
8.  $P_o(T)/g$ decreases from near zero to $P_{o-}/g$ (< 0) for $-6.00 \leq T \leq T_{Po-}$ or from $P_{o+}/g$ ($\approx -P_{o-}/g > 0$) to about zero for $T_{Po+} \leq T \leq 6.00$, exhibiting an increase in the favorable pressure gradient or decrease in the adverse pressure gradient in the FSZ. Moreover, it increases from $P_{o-}/g$, via 0, to $P_{o+}/g$ for $T_{Po-} \leq T < 0$, $T = 0$, and $0 < T \leq T_{Po+}$, indicating the change from favorable, via zero, to an adverse pressure gradient.
9.  With an increase in $H_0/h_0$, $P_{o+}/g$ increases but $T_{Po+}$ decreases. For $H_0/h_0$ = 0.179, 0.363 and 0.550, the values of $P_{o+}/g$ are about 5.74, 14.54 and 19.84 times that (= 0.0061) for $H_0/h_0$ = 0.050, showing the strong nonlinear effect on $P_o(T)/g$ and $P_{o+}/g$.

10. For each case, the incipient flow reversal occurs immediately after the maximum adverse pressure gradient. Namely, $T_{\mathrm{Po+}}$ is slightly less than $T_{\mathrm{ifr}}$. Further, $T_{\mathrm{ifr}}$ decreases if $H_0/h_0$ increases, which accentuates the nonlinear effect on the incipient flow reversal right above the bed.

**Supplementary Materials:** The following supporting information can be downloaded at: https://www.mdpi.com/article/10.3390/w14223609/s1, Herein, an original motion picture for Case E is supplemented as a visual aid to understand the flow structure in the near-bottom zone.

**Author Contributions:** C.L. was responsible for project administration, technical supervision and quality control of experimental results, and funding acquisition. Methodology, image processing, and software were handled by, M.-J.K. and S.-C.H.; The manuscript was written by C.L. Manuscript modifications and corrections were completed by J.Y. and J.-M.Y. All authors have read and agreed to the published version of the manuscript.

**Funding:** This research was supported by the Ministry of Science and Technology, Taiwan via Grant Nos. MOST 106-2221-E-005-045-MY3, MOST 108-2221-E-005-015-MY3 and MOST 109-2221-E-005-026-MY3 to Department of Civil Engineering, National Chung Hsing University, Taichung, Taiwan; and MOST 108-2115-M-126-003 and MOST 109-2115-M-126-002 to Department of Data Science and Big Data Analytics, Providence University, Taichung, Taiwan. This study was a collaboration with the KTH Royal Institute of Technology, Stockholm, Sweden.

**Data Availability Statement:** The data that support the findings of this study are available from the corresponding author upon reasonable request.

**Acknowledgments:** The authors are thankful to Po-Yu Chuang, Jie-Ming Syu, and Wei-Chih Pan for conducting HSPIV measurements and data analysis. Many thanks also go to UTOPIA Instruments Co., Ltd. for providing the high-speed, high-resolution digital Phantom camera, VEO640.

**Conflicts of Interest:** The authors declare no conflict of interest.

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
