# Peer review of "Effects of Nonlinearity on Velocity, Acceleration and Pressure Gradient in Free-Stream Zone of Solitary Wave over Horizontal Bed—An Experimental Study"

_water, doi:10.3390/w14223609_

Round 1

Reviewer 1 Report

The paper presentation is well and experimental results will be sound and these results will be referenced for several numerical studies.

- First, the authors should add several validation tests.

- Second,  the authors should add numerical simulations for their results. 

Many numerical simulations for wave paddle of solitary wave are done, you can cite those studies like:

‎Incompressible ‎smoothed particle hydrodynamics simulations on free surface flows AM Aly

International Journal of Industrial Mathematics (IJIM) 7 (1), 99-    

- Third, the quality of Fig. 12 is not clear.  It should be modified. 

Reviewer 2 Report

1. Abstract needs to be revised to include brief on results and investigated parameters.

2. Introduction section needs to be expanded to include discussion on prediction of the free surface elevation and kinematic features of a solitary wave traveling on a horizontal bed.

3. Research objectives are missing from the paper.

4. Include the uncertainty analysis of the HPIV technique.

5. Description on wave flume generation should have been included. Include the wave maker set-up.

6. There are numerous occasions in the paper that referencing to your previous publications of [15,17,18,21,22,24–26] - what is most significant difference of the current paper to those?

7. Time-dependent results are mostly discussed. Averaged quantities should have been provided as well for comparison.

8. Discuss how the non-dimensional quantities are obtained.

9. Figure 12 needs a better resolution (focused images).

10. How boundary layers are determined?
